# FreeCG: Free the Design Space of Clebsch–Gordan Transform for Machine Learning Force Fields

**Shihao Shao**[*][†]**, Haoran Geng**[‡]**, Zun Wang**[§]** & Qinghua Cui**[*][†]

## Abstract

Machine Learning Force Fields (MLFFs) are of great importance for chemistry, physics, materials science, and many other related fields. The Clebsch–Gordan transform (CG transform) effectively encodes many-body interactions and is thus an important building block for many models of MLFFs. However, the permutation-equivariance requirement of MLFFs limits the design space of CG transform, that is, intensive CG transform has to be conducted for each neighboring edge and the operations should be performed in the same manner for all edges. Freeing up the design space can greatly improve the model's expressiveness while simultaneously decreasing computational demands. To reach this goal, we utilize a mathematical proposition, invariance transitivity, to show that implementing the CG transform layer on the permutation-invariant abstract edges allows complete freedom in the design of the layer without compromising the overall permutation equivariance. Developing on this free design space, we further propose group CG transform with sparse path, abstract edges shuffling, and attention enhancer to form a powerful and efficient CG transform layer. Our method, known as *FreeCG*, achieves state-of-the-art (SOTA) results in force prediction for MD17, rMD17, MD22, and is well extended to property prediction in QM9 datasets with several improvements greater than 15% and the maximum beyond 20%. The extensive real-world applications showcase high practicality. FreeCG introduces a novel paradigm for carrying out efficient and expressive CG transform in future geometric network designs. To demonstrate this, the recent SOTA, QuinNet, is also enhanced under our paradigm. Code: https://github.com/ShihaoShao-GH/FreeCG.

## 1 Introduction

Machine Learning Force Fields (MLFFs) are of great importance for drug development (Chen et al., 2024), materials science (Liu et al., 2024), chemical reaction kinetics (Meuwly, 2021), nanotechnology (Wang et al., 2023b), among others. It offers a satisfactory trade-off between accuracy and efficiency, which is expected to perform as powerful as Density Functional Theory (DFT) (Kohn & Sham, 1965) or other high accuracy references (Martin, 2020; Ceperley & Alder, 1980; Bartlett & Musiał, 2007), but with orders-of-magnitude speedup (Cui et al., 2024; Wang et al., 2023c; 2024; Musaelian et al., 2023; Batzner et al., 2022; Drautz, 2019; Batatia et al., 2022b; Thölke & De Fabritiis, 2021; Schütt et al., 2018; Chmiela et al., 2017).

Graph Neural Networks (GNNs) perform SOTA on several MLFFs benchmarks (Schütt et al., 2018; 2021). Group and group representation theory play important roles in the design of GNNs for MLFFs. For instance, rotation invariance is generally required in these works, as we naturally require the potential energy unchanged w.r.t. rotations of the input molecule. A recent design trend is to maintain rotation, reflection, and translation equivariance in the design of neural networks. They hope the internal features can move with respect to the input molecule, enabling higher expressive power. GNNs that obey this property are called Equivariant Graph Neural Networks (EGNNs)

---

[*]Corresponding authors (shaoshihao@pku.edu.cn; cuiqinghua@hsc.pku.edu.cn)

[†]School of Basic Medical Sciences, Peking University

[‡]EECS, UC Berkeley

[§]AI4Science, Microsoft

(Thomas et al., 2018; Satorras et al., 2021; Gasteiger et al., 2020b; Vaswani et al., 2017; Fuchs et al., 2020; Liao & Smidt, 2022). To better model many-body interactions, irreducible representations (irreps) are adopted to represent high-order geometric objects. In this context, the Clebsch-Gordan (CG) transform is used to translate between different irreps. Several works leverage such high degree irreps or tensors, showing significant performance boost (Batatia et al., 2022b; Batzner et al., 2022; Musaelian et al., 2023; Gasteiger et al., 2021; Thomas et al., 2018; Simeon & De Fabritiis, 2024). However, the benefit of high degree irreps and CG transform performed on them is at the cost of heavy computational overhead. The reason is, tensors are extensions of scalars and vectors, and in this way CG transform also extends the dot product. Thus, the higher the degree of irreps for the CG transform, the greater the computational demands. The requirements for being permutation equivariant make this burden hard to alleviate. Unlike rotation or translation equivariance, permutation equivariance is often implicitly guaranteed in EGNNs, which means the order of internal features should changes according to it of input atoms. To maintain permutation equivariance, EGNNs require each node to receive information from neighboring atoms together with the edges linking them, where the heavy computation of CG transform occurs for each neighboring atom and edge. This means we cannot naïvely remove some neighbor computations, as it will break permutation equivariance. Moreover, the narrowness of the design space prevents us from freely constructing the CG transform layer, and thus limits the expressivity of models. For instance, we need to operate on each neighboring atom in an equal way (*e.g.,* the predecessors typically assign a same MLP operating on scalar features of the edge to produce the weights for each computation between the central atom and each neighboring one (Batzner et al., 2022; Musaelian et al., 2023)).

In this work, to confront this challenge, we propose *FreeCG*. The model generates and refines geometric features from the surrounding edges near each atom. We call the different aggregated edge geometric features *abstract edges*, which are permutation invariant when we consider the internal features maintained by a given atom w.r.t. the neighbouring atoms and edges. By the invariance transitivity, we show that CG transform on these abstract edges is also permutation invariant, regardless of designs, and does not affect the permutation equivariance of the layer, thus being free of the burdens above. Furthermore, the abstract edges are constructed from different real edges, so they contain refined features of them for better model expressive power. The invariance nature of abstract edges allows us to assign different weights to different edges, instead of weights computed by the same MLP. The free design space allows us to do more. We put abstract edges into groups, and operate on each group individually, to further decrease the computation demands. Previous works that keep E(3)-equivariance are more expensive (Batzner et al., 2022; Musaelian et al., 2023), as they require an extra *parity* argument being $1$ or $-1$, and thus the number of irreps is doubled. Instead, we select an efficient set of paths for CG transform so that we maintain E(3)-equivariance while being more efficient than keeping SE(3)-equivariance. The abstract edges shuffling, inspired by (Zhang et al., 2018), is also implied for combination of irreps features. The abstract edges are then plugged back into the cross-attention calculation to improve the quality of the attention scores. The operations above are available thanks to the invariance properties of the abstract edges.

The contributions are summarized as follows:

**1)** We utilize the invariance transitivity with permutation-invariant abstract edges to resolve a major challenge in the current EGNNs: the narrowness design space of CG transform, which results in reduced expressivity and high computation overhead.

**2)** We propose *FreeCG*, comprising of three main components: ***Group CG transform with sparse path***, ***abstract edges shuffling***, and ***Attention enhancer***. These contribute to an informative and efficient model with high-order irreps and CG transform.

**3)** Experiments on small molecule datasets MD17, rMD17, large molecules ones MD22, and molecular property datasets QM9 reveal the SOTA performance of FreeCG, with several improvements beyond 15%. The extensive real-world applications further indicate the practicality.

**4)** This work presents a new paradigm for CG transform in future research, extending beyond the design presented here. We further enhance QuinNet (Wang et al., 2023c) under this paradigm to demonstrate this point.

## 2 RELATED WORKS

Maintaining E(3)-/SE(3)-equivariance has become a popular trend in the design of neural networks for MLFFs. In general, the methods can be categorized into two lines:

**Methods with geometric vectors.** Several methods directly work on regular geometric vectors (Schütt et al., 2021; 2018; Gasteiger et al., 2020b;a). The starting point of this line of works is about covering necessary information for MLFFs. For instance, SchNet (Schütt et al., 2018) first introduces continuous-filter to MLFFs, but only distance information is covered. DimeNet (Gasteiger et al., 2020b) further considers bond angle information to obtain higher capacity. To make models capable of capturing many-body interactions, most of the following models in this line explicitly encode such types of interactions (Thölke & De Fabritiis, 2021; Wang et al., 2024; 2023c). This is commonly done via calculating different angles between atoms or surfaces (e.g., torsion angle, improper angle (Wang et al., 2024), dihedral angle (Wang et al., 2023c)). The irreducible representations (irreps) and Clebsch-Gordan (CG) transform, on the other hand, can implicitly encode many-body interactions in well-defined mathematical objects. This work presents FreeCG with highly efficient and expressive CG transform layers, significantly outperforming previous works by maximum margins and with minor computational load.

**Methods equipped with irreps and CG transform.** Applying irreps and CG transform to Equivariant Graph Neural Networks (EGNNs) was first proposed as more of a conceptual framework, known as Tensor Field Network (Thomas et al., 2018). Several works have been proposed on top of this foundation. For instance, NequIP (Batzner et al., 2022) has implemented the idea to construct high-order irreps, and shows state-of-the-art (SOTA) results for MLFFs. Allegro (Musaelian et al., 2023) resolves the challenges of the scaling issue, which makes it possible to run parallelly on a large number of GPUs. Furthermore, Allegro constructs high order geometric features with pair-wise edge CG transform, which is similiar to FreeCG at this point. However, Allegro requires to compute the same CG transform for each pair of edges in the same manner to maintain permutation equivariance, while FreeCG is capable of designing different CG transform for each pair of abstract edges because of their permutation invariance. The atom-centric style and the rich representation of abstract edges together makes stronger expressivity of FreeCG. SE(3)-Transformer (Fuchs et al., 2020) first proposes equivariant dot-product for generating self-attention. Equiformer (Liao & Smidt, 2022) further achieves E(3)-equivariance combining MLP attention and non-linear messages. MACE (Batatia et al., 2022b) extends classic body order expansion methods, Atomic Cluster Expansion (ACE) (Drautz, 2019), to a hierarchical framework. CG transform is a fundamental building block in these works, but with a limited design space, affecting both performance and efficiency. In this work, we completely free the design space of CG transform and propose a novel model, FreeCG, performing strong SOTA and showing high efficiency for MLFFs, which is also a new design paradigm for future works.

## 3 METHODS

### 3.1 BACKGROUND

**Group, equivariance and invariance.** Permutation, rotation, and translation form different groups in group theory. Formally, a set with a binary operation $(G, *)$ is said to be a group if and only if the following conditions hold: 1) $g_1 * g_2 \in G$, for any $g_1, g_2 \in G$ (closure) 2) $(g_1 * g_2) * g_3 = g_1 * (g_2 * g_3)$, for any $g_1, g_2, g_3 \in G$ (associativity) 3) There exists a group element $e \in G$, such that $g * e = e * g = g$, for any $g \in G$. ($e$ identity element) 4) There is a group element $g'$ w.r.t. $g$, such that $g * g' = g' * g = e$, for each $g \in G$ ($g'$ inverse element). The group elements $g \in G$, according to the representation theory, can be represented as linear transformations $\mathcal{P}_V(g) \in GL(V)$ on vector space $V$. Given a function $f : X \to Y$, where $X$ and $Y$ are vector spaces. It is said to be $G$-equivariant if and only if $f(\mathcal{P}_X(g)x) = \mathcal{P}_Y(g)f(x)$, for any $g \in G$. $G$-invariance is a special case when $\mathcal{P}_Y(g)$ is an identity matrix. Permutation equivariance and E(3)-equivariance are two properties each layer of our model obeys. Permutation equivariance means the index of node or edge features will be consistent when passing a layer. E(3)-equivariance covers rotation, translation, and reflection, where the translation is explicitly guaranteed via only considering the relative distances between atoms, thus we consider O(3)-equivariance where translations are omitted. It is intuitive to correspondingly change directional features when the whole molecule rotates or reflects.

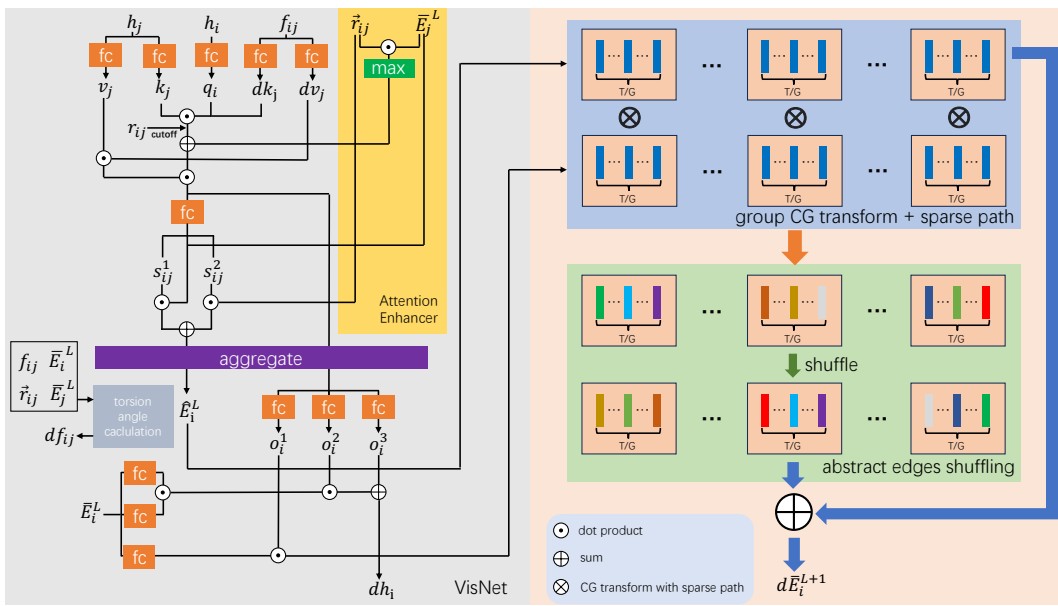

Figure 1: The architecture of a single layer of FreeCG. The cross-attention mechanism generates abstract edges through a permutation-invariant process. The abstract edges are also used to enhance the quality of the attention score, denoted as *Attention Enhancer*. In the right part, the *Group CG transform* organizes abstract edges into groups and performs the CG transform on each group. We adopt *sparse path* for CG transform, enabling lower computation demands while maintaining O(3) equivariance. *Abstract edges shuffling* improves the information exchange between different irreps. Details for sparse path and abstract edges shuffling can be referred to Fig. 2. Better viewed in color.

**Tensor, irreps and CG transform.** Tensors are high-dimensional generalizations of scalars, vectors, and matrices. Scalars and vectors are both special cases of Cartesian tensors. Tensor product can generate high-rank tensors from low-rank ones. Formally, tensors are the results of tensor product of several vectors and covectors. In our context, it is not essential to distinguish between vectors and covectors. Tensors representing groups can be further decomposed to the direct sum of irreps. For example, tensors of SO(3) (omit reflection compared to O(3)) on 9-space (from tensor product of two $3 \times 3$ rotation matrix) can be decomposed into $1 \times 1$ ($l = 0$), $3 \times 3$ ($l = 1$), and $5 \times 5$ ($l = 2$) irreps, which are called Wigner-D matrices. In EGNNs, we often project the distance vector between atoms onto the unit sphere $S^2$ with the central atom as the center of sphere. Actually, $S^2$ is homomorphic to the quotient group SO(3)/SO(2), thus it also has its own irreps, *e.g.*, $l = 0$ scalar and $l = 1$ vector. $S^2$ irreps are the main features we maintain in our model, where irreps with degree $l$ has $2l + 1$ elements, which are often indexed by $m$. To combine these features, we can calculate the tensor product between them, and the results can, again, be decomposed to irreps. This process is known as CG transform, which utilizes CG coefficients to perform transformations. For instance, $A_{m_1}^{1,l_2 l_3 \mapsto l_1} = \sum_{m_2,m_3} C_{m_1 m_2 m_3}^{l_1 l_2 l_3} A_{m_2}^{2,l_2} A_{m_3}^{3,l_3}$, where $A^l$ are $S^2$ irreps, $m$ denotes the elements of irreps, and $C$ the CG coefficient. To satisfy O(3), we consider an additional variable, parity $p$, which takes the values of 1 or $-1$. Irreps with $p = -1$ will be inverse when the space is reflected, and $p = 1$ unchanged. The above formula of CG transform becomes:

$$A_{m_1}^{1,l_2 p_2 l_3 p_3 \mapsto l_1 p_1} = \mathbb{1}_{(p_1 = p_2 p_3)} \sum_{m_2,m_3} C_{m_1 m_2 m_3}^{l_1 l_2 l_3} A_{m_2}^{2,l_2 p_2} A_{m_3}^{3,l_3 p_3} \qquad (1)$$

where $\mathbb{1}_{(expression)}$ is the indicator function, outputting 1 if $expression$ is true, and 0 otherwise. Given a vector ($l = 1$ $S^2$ irreps), we can *lift* it to irreps with arbitrary degree $l$ and $p = (-1)^l$, via a series of real spherical harmonics ($Y_{m=1}^l, ..., Y_{m=2l+1}^l$). For further details about group theory, we refer interested readers to related books and papers (Zee, 2016; Raczka & Barut, 1986; Thomas et al., 2018; Jeevanjee, 2011; Cohen et al., 2018).

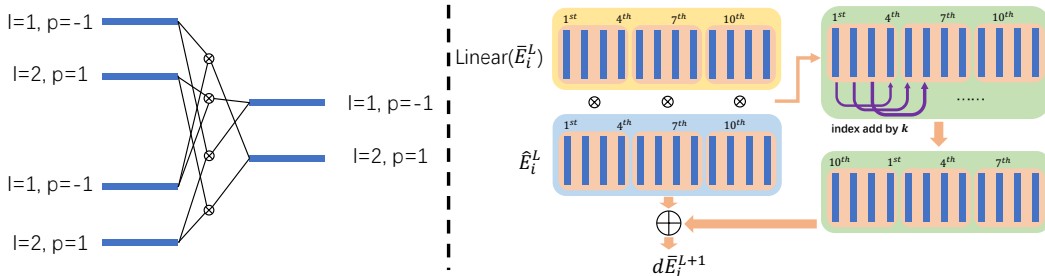

Figure 2: Details on sparse path and abstract edges shuffling. **Left:** The sparse path holds two useful properties: 1) The number of paths is less than the weaker SO(3) equivariance (4 vs. 8). 2) Each output irreps contains the information from input ones with both degree $l = 1$ and $l = 2$. **Right:** The shuffling strategy is to add a constant $k$ for the index of each abstract edge. The shuffled result is then added by $\hat{E}_i^L$, and get the final added value $d\overline{E}_i^{L+1}$. Better viewed in color.

## 3.2 PROBLEM ANALYSIS

The task of force field prediction can be formalised as follows: Given a set of atoms with their positions and atom types $\{X, Z\}$, the neural network $f_\theta$ with parameter $\theta$ aims to predict the energy, and by which it derives the predicted force on each atom. In each layer of NequIP (Batzner et al., 2022), messages from neighboring atoms are aggregated and combined with the features of the central atom. The messages are created via CG transform between the irreps. Here, we revisit the critical step constructing messages to a central atom $a$ in NequIP:

$$
\mathcal{L}_{acm_o}^{l_e p_e l_n p_n \mapsto l_o p_o}(X, N) = \mathbb{1}_{(p_o = p_e p_n)} \sum_{m_e m_n} C_{m_o m_e m_n}^{l_o l_e l_n}
$$
$$
\sum_{b \in \mathcal{N}(a)} (R(\|\vec{r}_{ab}\|)_c^{l_o l_e l_n}) Y_{m_e}^{l_e}(\frac{\vec{r}_{ab}}{\|\vec{r}_{ab}\|}) N_{bcm_n}^{l_n p_n}
$$

(2)

where $\mathcal{N}(a)$ is the set of neighboring atoms of atom $a$. $R$ is a MLP. $\|*\|$ is Euclidean norm. $N_b$ is the features of node $b$. $\vec{r}_{ab}$ is the vector pointing from atom $a$ to $b$. Consider the vector function form of Eq. 2: $\mathcal{L}_{cm_o}^{l_e p_e l_n p_n \mapsto l_o p_o} = (\mathcal{L}_{1cm_o}^{l_e p_e l_n p_n \mapsto l_o p_o}, \mathcal{L}_{2cm_o}^{l_e p_e l_n p_n \mapsto l_o p_o}, ...)$, which is permutation equivariant w.r.t. permutation operations acting on $X$ and $Z$. Formally, it means $\mathcal{L}_{cm_o}^{l_e p_e l_n p_n \mapsto l_o p_o}(\mathcal{P}_X X, \mathcal{P}_Z Z) = \mathcal{P}_{\mathcal{L}} \mathcal{L}_{cm_o}^{l_e p_e l_n p_n \mapsto l_o p_o}(X, Z)$. Put simply, if we exchange the *indexes* of two atoms, for example, 1 and 2, and feed them into function $\mathcal{L}_{cm_o}^{l_e p_e l_n p_n \mapsto l_o p_o}$, it equals to that we directly change the index 1 and 2 of the output of function $\mathcal{L}_{cm_o}^{l_e p_e l_n p_n \mapsto l_o p_o}$, which is $(\mathcal{L}_{2cm_o}^{l_e p_e l_n p_n \mapsto l_o p_o}, \mathcal{L}_{1cm_o}^{l_e p_e l_n p_n \mapsto l_o p_o}, ...)$. This property is simple and very important for the molecular neural networks, as the indexes of atomic features should change correspondingly w.r.t. the changes of atom indexes, and the feature values for each atom should stay unchanged.

Most works take this property for granted. However, the permutation equivariance is actually important but vulnerable. It limits the design space to a very small scope, and make the network poorly scalable when the number of neighbors arises. Specifically, it brings the following issues:

**Problem 1** *The CG transform layer scales as $\mathcal{O}(\sum_i \text{card}(\mathcal{N}(i)))$, where $\text{card}(X)$ is the number of elements in set $X$. One cannot arbitrarily remove calculations for a specific neighboring atom because it would break the permutation equivariance.*

**Problem 2** *The design space is limited for maintaining permutation equivariance. For example, in Eq. 2, the formulation and the parameters of $R$ should be the same across different neighboring atoms, thus forbidding the design for complicated CG transform layers.*

Problem 1 poses heavy computation challenges, as the CG transform itself is very time-consuming, compared to dot product and element-wise multiplication. We provide a detailed analysis for the efficiency of CG transform in Sec. A.6. On the other hand, the narrowness for design space brought by problem 2 makes it hard to design a high expressive CG transform layer, as only limited structures can be designed to maintain permutation equivariance. To address these problems, we aim to

free the CG transform in messages transmissions from the constraints of permutation equivariance without compromising the overall equivariance of the network. Here, we leverage a simple and useful mathematical property. Consider a function $h$ that can be written as:

$$h(x) = h^{'}(h_1(x), h_2(x), ...) \tag{3}$$

if $h_*(x)$ are all $G$-invariant, then, regardless of how we design $h^{'}$, the overall function $h$ must be $G$-invariant as well. The proof is simple, as:

$$h^{'}(h_1(\mathcal{P}_X(g)x), h_2(\mathcal{P}_X(g)x), ...) = \mathcal{P}_h(e)h^{'}(h_1(x), h_2(x), ...) \tag{4}$$

Here, the invariance we study is about the internal features of a given atom w.r.t. the neighbours. Specifically, it is the sum $\sum_{b \in \mathcal{N}(a)} (R(\|\vec{r}_{ab}\|)_c^{l_o l_e l_n}) Y_{m_e}^{l_e}(\frac{\vec{r}_{ab}}{\|\vec{r}_{ab}\|}) N_{bcm_n}^{l_n p_n}$, and the term for each $b$. Such invariant components guarantee the equivariance of the layer mentioned above. Thus, we can freely design the function $h^{'}$ once we have these invariant functions $h_*$.

### 3.3 FreeCG

**Abstract edges.** The above proposition presents an elegant way to solve problem 1 and 2. The idea is to put CG transform inside the function $h^{'}$, and by the conclusion, we can completely free the design space of the CG transform. The first step is to construct the permutation invariant function $h_*$. To emphasize the geometric information, we want these $h_*$'s to be the aggregation of edge features. We call $h_*$'s *abstract edges*. For the concrete design, we take the transformer architecture in ViSNet (Wang et al., 2024) as an efficient tool to construct abstract edges. The detailed information of the complete implementation is in the Sec. A.2. In ViSNet, each edge maintains high-degree features $E_{ij} = E_{ij}^{l=1} \oplus E_{ij}^{l=2}$ consisting of irreps $E_{ij}^l = Y^l(\vec{r}_{ij}/\|\vec{r}_{ij}\|)$. The above features are invariant to layer index $L$. The computed attention $a_{ij}^{L,t}$ is multiplied to each edge. The sum of them $\hat{E}_{i,t}^L = \sum_{ij \in \mathcal{E}(i)} a_{ij,t}^L E_{ij}$ forms a temporary abstract edge, where we omit the degree $l$, and $L$ the index of the layer. $t$ denotes the index of the $t$-th abstract edges ($\hat{E}_{i,t=1}^L, \hat{E}_{i,t=2}^L, ...$). In the original ViSNet, it was used to update the geometric feature $d\overline{E}_i^{L+1} = \hat{E}_i^L + o_i^{L,1} \cdot \text{Linear}(\overline{E}_i^L)$, where Linear is a fully-connected linear operation, which performs across the dimension of $t$, thus does not break the equivariance, and $\overline{E}$ represents propagated abstract edges in contrast to the temporary one $\hat{E}$, where the former is updated by $d\overline{E}$ in each layer, derived from the temporary abstract edge $\hat{E}$. $o_i^{L,1}$ is a variable generated from the node feature, as we will introduce in Sec. A.2. We utilize that $\hat{E}^L$ and $\overline{E}^L$ fits the requirements of $h_*$ in our proposition, take them as abstract edges, and propose methods to construct CG transform function $h'$ upon it. The proof that each abstract edge meets the requirement for $h_*$, namely, it is permutation invariant, is in Sec. A.4.

**Group CG transform.** The number of abstract edges is decided by us, so the complexity for computing CG transform in Problem 1 is controlled to be constant. The proposition above gives us complete freedom to construct the CG transform function $h'$, expanding the design space to maximum, alleviating Problem 2. The idea is to use CG transform function $h'$ to replace the updating mechanism of $\overline{E}$ in ViSNet. A naïve attempt is to directly take the CG transform between $\overline{E}^L$ and $\hat{E}^L$ to acquire $\overline{E}^{L+1}$. However, we want to further decrease the $O(T^2)$ time complexity for the CG transform, where $T$ is the number of abstract edges, even though it is a constant number. Leveraging the unlimited freedom in constructing $h$, and taking inspiration of group convolution (Krizhevsky et al., 2012), we propose group CG transform (distinct from the group in group theory). We first split the abstract edges of $\overline{E}^L$ and $\hat{E}^L$ into groups, where each index of abstract edge belongs to some group $U_g$, the integer $g$ ranges from 1 to $G$, and $G$ a hyper-parameter for the number of total groups. Then a group CG transform acts as:

$$d\overline{E}_{i,t_o m_o}^{'L+1,l_o,p_o} = \mathbb{1}_{(p_o=p_1 p_2)} o^{L,1} \sum_{l_1,l_2} \sum_{m_1,m_2} C_{m_o m_1 m_2}^{l_o,l_1,l_2}$$
$$\sum_{t_1,t_2 \in U_g} W_{t_o t_1 t_2}^{l_o,l_1,l_2} \text{Linear}(\overline{E}_i^L)_{t_1 m_1}^{l_1,p_1} \hat{E}_{i,t_2 m_2}^{L,l_2,p_2} \tag{5}$$

Table 1: Performances on MD17 dataset. The results are reported in mean absoulute error (MAE). The energies and forces are measured in kcal/mol and kcal/mol/Å, respectively. The best numbers are marked in **bold**.

| Molecule | SchNet | DimeNet | PaiNN | SpookeyNet | ET | GemNet | NequIP | SO3KRATES | ViSNet | QuinNet | FreeCG |
|---|---|---|---|---|---|---|---|---|---|---|---|
| | | | | | *Energy Prediction* | | | | | | |
| Aspirin | 0.37 | 0.204 | 0.167 | 0.151 | 0.123 | - | 0.131 | 0.139 | 0.116 | 0.119 | **0.110** |
| Ethanol | 0.08 | 0.064 | 0.064 | 0.052 | 0.052 | - | 0.051 | 0.052 | 0.051 | 0.050 | **0.049** |
| Malondialdehyde | 0.13 | 0.104 | 0.091 | 0.079 | 0.077 | - | 0.076 | 0.077 | **0.075** | 0.078 | 0.094 |
| Naphthalene | 0.16 | 0.122 | 0.116 | 0.116 | 0.085 | - | 0.113 | 0.115 | 0.085 | 0.101 | **0.083** |
| Salicylic acid | 0.20 | 0.134 | 0.116 | 0.114 | 0.093 | - | 0.106 | 0.016 | 0.092 | 0.101 | **0.090** |
| Toluene | 0.12 | 0.102 | 0.095 | 0.094 | **0.074** | - | 0.092 | 0.095 | **0.074** | 0.080 | 0.076 |
| Uracil | 0.14 | 0.115 | 0.106 | 0.105 | **0.095** | - | 0.104 | 0.103 | **0.095** | 0.096 | 0.097 |
| | | | | | *Force Prediction* | | | | | | |
| Aspirin | 1.35 | 0.499 | 0.338 | 0.258 | 0.253 | 0.217 | 0.184 | 0.236 | 0.155 | 0.145 | **0.122** |
| Ethanol | 0.39 | 0.230 | 0.224 | 0.094 | 0.109 | 0.085 | 0.071 | 0.096 | 0.060 | 0.060 | **0.053** |
| Malondialdehyde | 0.66 | 0.383 | 0.319 | 0.167 | 0.169 | 0.155 | 0.129 | 0.147 | 0.100 | 0.097 | **0.095** |
| Naphthalene | 0.58 | 0.215 | 0.077 | 0.089 | 0.061 | 0.051 | 0.039 | 0.074 | 0.039 | 0.039 | **0.034** |
| Salicylic acid | 0.85 | 0.374 | 0.195 | 0.180 | 0.129 | 0.125 | 0.090 | 0.145 | 0.084 | 0.080 | **0.070** |
| Toluene | 0.57 | 0.216 | 0.094 | 0.087 | 0.067 | 0.060 | 0.046 | 0.073 | 0.039 | 0.039 | **0.035** |
| Uracil | 0.56 | 0.301 | 0.139 | 0.119 | 0.095 | 0.097 | 0.076 | 0.111 | 0.062 | 0.062 | **0.059** |

where $t_o \in U_g$. The group CG transform decreases the time complexity to $O(T^2/G)$. Here, the parameters $W$ for CG transform are also worth emphasizing. They are not necessary to be kept the same across different abstract edges $t$ to keep permutation equivariance, and do not need to adopt the same MLP for each edge to calculate weights. Thus, we directly assign different weights $W$ for different abstract edges to simplify the model design. In contrast to previous methods, We save the computational cost for calculating weights for each edge.

**Sparse path.** Typically, ensuring SO(3) equivariance is considered more effcient than ensuring O(3) equivariance. It is because we often need to consider both $p = 1$ and $p = -1$ for a single $l$ for O(3), thus the total computation is quadrupled, and memory usage is doubled. Here we propose a method to keep O(3) while being more efficient than SO(3). We only keep $(l = 1, p = -1)$ and $(l = 2, p = 1)$, which is same as the order of directly using spherical harmonics. In such way, It suffices that each output irreps containing information from both input irreps through CG transform, as illustrated in Fig. 2. There are only 4 path in contrast to 8 path for SO(3), being O(3) equivariant but more efficient than being SO(3) equivariant.

**Abstract edges shuffling.** Inspired by ShuffleNet (Zhang et al., 2018), we can also shuffle the abstract edges to make the information exchanged comprehensively. If we omit the shuffling operation, the information in the abstract edges would only be exchanged within the group, which decreases the capacity of the model. We shuffle all the abstract edges. Specifically, we increase the indices of all irreps by $\lfloor 1.5 * T/G \rfloor$. If the index exceeds $T$, we start counting from 1 again. Theoretically, the shuffling strategy can be arbitrary as long as maintaining the same strategy for each layer during every inference. This process is also depicted in Fig. 2. The ablation on different strategies is shown in Sec. 4.4.

**Abstract edges enhance cross-attention.** The transformer integrates neighboring atoms information in the model through cross-attention mechanism, which aims to capture relations for those atoms exhibiting strong interatomic correlations. Thanks to the informative abstract edge, we utilize it to augment the generation of attention scores. In the original design, to calculate the cross-attention, the node scalar features are processed to generate query $Q$, key $K$, and value $V$ for each atom, respectively. Then, the self attention is computed as $a_{ij} = \text{SiLU}(\text{Cutoff}(\|\vec{r}_{ij}\|)q_i k_j dk_j)$. Note that ViSNet is different from other transformer-based models where $a_{ij}$ is scaled by the SiLU instead of Softmax across different $j$. We integrate the information of abstract edges by:

$$a_{ij} = \text{SiLU}\left(\text{Cutoff}(\|\vec{r}_{ij}\|)q_i k_j dk_j + \text{AttEnhancer}(E_{ij}, \overline{E}_j^L)\right) \quad (6)$$

where $\text{AttEnhancer}(E_{ij}, \overline{E}_j^L) = \max_t(\overline{E}_{j,t}^L \odot E_{ij})$, that is, for each real edge feature $E_{ij}$, we compute the dot product with all abstract edges $\overline{E}_{j,t}^L$ and take the maximum value across different

Table 2: Performances on rMD17 dataset. The results are reported in MAE. The energies and forces are measured in kcal/mol and kcal/mol/Å, respectively. The best numbers are marked in **bold**.

| Molecule | UNiTE | GemNet | NequIP | MACE | Allegro | BOTNet | ViSNet | QuinNet | ACE$^g$ | PΘNITA | FreeCG |
|---|---|---|---|---|---|---|---|---|---|---|---|
| | | | | | Energy Prediction | | | | | | |
| Aspirin | 0.055 | - | 0.0530 | 0.0507 | 0.0530 | 0.0530 | 0.0445 | 0.0486 | **0.0392** | **0.0392** | 0.0530 |
| Azobenzene | 0.025 | - | 0.0161 | 0.0277 | 0.0277 | 0.0161 | 0.0156 | 0.0394 | **0.0115** | 0.0161 | 0.0217 |
| Benzene | 0.002 | - | 0.0009 | 0.0092 | 0.0069 | 0.0007 | 0.0007 | 0.0096 | **0.0002** | 0.0039 | 0.0107 |
| Ethanol | 0.014 | - | 0.0092 | **0.0032** | 0.0092 | 0.0092 | 0.0078 | 0.0096 | 0.0046 | 0.0592 | 0.0087 |
| Malonaldehyde | 0.025 | - | 0.0184 | 0.0185 | 0.0138 | 0.185 | **0.0132** | 0.0168 | 0.0115 | 0.0138 | 0.0146 |
| Naphthalene | 0.011 | - | **0.0046** | 0.1153 | **0.0046** | **0.0046** | 0.0057 | 0.0174 | **0.0046** | 0.0069 | 0.0118 |
| Paracetamol | 0.044 | - | 0.0323 | 0.0300 | 0.0346 | 0.0300 | 0.0258 | 0.0362 | **0.0208** | 0.0254 | 0.0392 |
| Salicylic acid | 0.017 | - | 0.0161 | 0.0208 | 0.0208 | 0.0185 | 0.0161 | 0.033 | **0.0115** | 0.0161 | 0.0233 |
| Toluene | 0.010 | - | 0.0069 | 0.0115 | 0.0092 | 0.0069 | 0.0059 | 0.0139 | **0.0046** | 0.0069 | 0.0334 |
| Uracil | 0.013 | - | 0.0092 | 0.0115 | 0.0138 | 0.0092 | 0.0069 | 0.0149 | **0.0046** | 0.0092 | 0.0116 |
| | | | | | Force Prediction | | | | | | |
| Aspirin | 0.175 | 0.2191 | 0.1891 | 0.1522 | 0.1684 | 0.1900 | 0.1520 | 0.1429 | 0.1407 | 0.1338 | **0.1212** |
| Azobenzene | 0.097 | - | 0.0669 | 0.0692 | 0.0600 | 0.0761 | 0.0585 | 0.0513 | 0.0507 | 0.0530 | **0.0486** |
| Benzene | 0.017 | 0.0115 | 0.0069 | 0.0069 | 0.0046 | 0.0069 | 0.0056 | 0.0047 | **0.0023** | 0.0069 | 0.0056 |
| Ethanol | 0.085 | 0.083 | 0.0646 | 0.0484 | 0.0484 | 0.0738 | 0.0522 | 0.0516 | **0.0322** | 0.0577 | 0.0438 |
| Malonaldehyde | 0.152 | 0.1522 | 0.0118 | 0.0946 | 0.0830 | 0.1338 | 0.0893 | 0.0875 | **0.0715** | 0.0922 | 0.0802 |
| Naphthalene | 0.060 | 0.0438 | 0.0300 | 0.0369 | **0.0208** | 0.0415 | 0.0291 | 0.0242 | **0.0208** | 0.0300 | 0.0228 |
| Paracetamol | 0.164 | - | 0.1361 | 0.1107 | 0.1130 | 0.1338 | 0.1029 | 0.0979 | 0.0922 | 0.0922 | **0.0840** |
| Salicylic acid | 0.088 | 0.1222 | 0.0922 | 0.0715 | 0.0669 | 0.0992 | 0.0795 | 0.0771 | **0.0623** | 0.0761 | 0.0648 |
| Toluene | 0.058 | 0.0507 | 0.0369 | 0.0350 | 0.0415 | 0.0438 | 0.0264 | 0.0244 | 0.0254 | 0.0300 | **0.0239** |
| Uracil | 0.088 | 0.0876 | 0.0669 | 0.0484 | **0.0415** | 0.0738 | 0.0495 | 0.0487 | 0.0392 | 0.0553 | 0.0446 |

abstract edges. $E_{ij}$ does not have an $L$ superscript because these features remain constant across different layers. This as an additional contribution to the cross-attention, as it quantifies how well the abstract edges capture the information of the edge linking atoms $i$ and $j$. The detailed implementation, including the Cutoff function and $dk_j$ in the above formula, is introduced in Sec. A.2.

**Equivariance.** It is important to maintain equivariance for physics-informed tasks. Here, we summarize the overall idea of ensuring equivariance for FreeCG. There are two equivariances maintained in this work, namely O(3) (3D rotation and reflection) and permutation equivariance. We maintain O(3) equivariance via leveraging spherical irreps, CG transform, and the linear combinations on them. Each of the objects and operations above is O(3)-equivariant, and so does the combination of them. Specifically, the proof of linear combination is in Sec. A.5. The core challenge of this work lies in how to maintain permutation equivariance while generating abstract edges. It is shown, in Sec. 3.3, that we leverage invariance transitivity to maintain the permutation equivariance.

## 4 EXPERIMENTS

To evaluate the performance of FreeCG, we collect force field datasets, on which we compare our methods with other SOTA MLFFs. These datasets include small molecules dataset MD17 (Chmiela et al., 2017) with its revised version, rMD17 (Christensen & Von Lilienfeld, 2020), and large molecules dataset MD22 (Chmiela et al., 2023). To test the generalizability of FreeCG, we also evaluate the performance of FreeCG on a standard molecule property prediction dataset, QM9 (Ruddigkeit et al., 2012; Ramakrishnan et al., 2014). We take popular SOTA models into the comparison, including sGDML (Chmiela et al., 2017), SchNet (Schütt et al., 2018), DimeNet (Gasteiger et al., 2020b), SphereNet (Liu et al., 2021), PaxNet (Zhang et al., 2022), PaiNN (Schütt et al., 2021), SpookyNet (Unke et al., 2021), ForceNet (Hu et al., 2021), ET (Thölke & De Fabritiis, 2021), GemNet (Gasteiger et al., 2021), ComENet (Wang et al., 2022), NequIP (Batzner et al., 2022), UniTE (Qiao et al., 2022), SO3KRATES (Frank et al., 2022), MACE (Batatia et al., 2022b), Allegro (Musaelian et al., 2023), BOTNet (Batatia et al., 2022a), ViSNet (Wang et al., 2024), ViSNet-LSRM (Li et al., 2023), QuinNet (Wang et al., 2023c), ACE$^g$ (Bochkarev et al., 2024), PΘNITA (Bekkers et al., 2024),and EquiformerV2 (Liao et al., 2024). To asses the practicality on real-world tasks, Molecular dynamics simulations are run for MD17 molecules and two periodic systems, water (Fu et al., 2022; Wu et al., 2006) and LiPS (Batzner et al., 2022) under Periodic Boundary Conditions

Table 3: Performances on MD22 dataset. The results are reported in MAE. The energies and forces are measured in kcal/mol and kcal/mol/Å, respectively. The best numbers are marked in **bold**. Note that the energy MAE is calculated without being divided by the total number of atoms as Wang et al. (2024), unlike Wang et al. (2023c); Chmiela et al. (2017; 2023), which does not affect the comparison.

| Molecule | sGDML | ViSNet | ViSNet-Improper | ViSNet-LSRM | MACE | QuinNet | FreeCG |
|---|---|---|---|---|---|---|---|
| | | | *Energy Prediction* | | | | |
| Ac-Ala3-NHMe | 0.391 | 0.0636 | 0.0546 | 0.0673 | 0.0631 | 0.0840 | **0.0507** |
| AT-AT | 0.720 | 0.0708 | 0.0668 | 0.0780 | 0.108 | 0.144 | **0.0665** |
| AT-AT-CG-CG | 1.42 | 0.196 | 0.197 | **0.118** | 0.154 | 0.379 | 0.254 |
| DHA | 1.29 | 0.0741 | **0.0700** | 0.0897 | 0.135 | 0.118 | 0.0761 |
| Buckyball catcher | 1.17 | 0.508 | 0.537 | **0.319** | 0.489 | 0.563 | 0.512 |
| Stachyose | 4.00 | 0.0915 | 0.0882 | **0.104** | 0.122 | 0.226 | 0.183 |
| Double-walled nanotube | 4.00 | 0.800 | 0.601 | 1.81 | 1.67 | 1.81 | **0.543** |
| | | | *Force Prediction* | | | | |
| Ac-Ala3-NHMe | 0.790 | 0.0830 | 0.0709 | 0.0942 | 0.0876 | 0.0681 | **0.0531** |
| AT-AT | 0.690 | 0.0812 | 0.0776 | 0.0781 | 0.0992 | 0.0687 | **0.0634** |
| AT-AT-CG-CG | 0.700 | 0.148 | 0.139 | **0.1064** | 0.1153 | 0.1273 | 0.1252 |
| DHA | 0.750 | 0.0598 | 0.0554 | 0.0598 | 0.0646 | 0.0515 | **0.0507** |
| Buckyball catcher | 0.680 | 0.184 | 0.201 | 0.1026 | **0.0853** | 0.1091 | 0.1783 |
| Stachyose | 0.680 | 0.0879 | 0.0802 | 0.0767 | 0.0876 | **0.0543** | 0.612 |
| Double-walled nanotube | 0.520 | 0.362 | 0.292 | 0.3391 | 0.2767 | 0.2473 | **0.2449** |

(PBCs). FreeCG is also evaluated on the conformation space of a 166-atom mini-protein, Chignolin (Wang et al., 2023a). Mean Squared Error (MSE) loss is adopted for all experiments. The ablation on each component and hyper-parameters of FreeCG are also presented. The common settings and extra experiments are reported in the Sec. A.1.

## 4.1 COMPARISON WITH STATE-OF-THE-ARTS FOR MLFFs

**Force field on small molecules and periodic systems.** MD17 is a famous molecular dynamics benchmark for small molecules. FreeCG outperforms others in all force prediction tasks. It also significantly decreases the force prediction errors for the hardest-to-predict molecule in this datasets, aspirin, by 15%. Remarkably, FreeCG also decreases the MAE by over 10% for ethanol, naphthalene, and salicylic acid. FreeCG does not have a particular preference for the size of molecules. It demonstrates strong performance for aspirin (180.2 g/mol) and excels on ethanol (46.1 g/mol). The energy prediction is also competitive when compared to other SOTA methods. rMD17 is the revised version of MD17. It recomputed the trajectories of each atom with higher accuracy. The force prediction accuracy of FreeCG is still leading in majority of the molecules. It improves the force results compared to the baseline, ViSNet, in all the molecules except for benzene, and performs SOTA on more atoms. Note that the results on benzene is already extreme high with previous models. The results for MD17 and rMD17 can be referred to Tab. 1 and 2, respectively. The accuracy of force prediction on periodic systems, water and LiPS, are also evaluated. Fig. 8 and 9 show that, FreeCG achieves the best performance across other methods, and decrease the MAE value of the second best by 50% for water. Note that the energy prediction on rMD17 does not show SOTA results. The main focus of our work is for force prediction, and the downstream task is mainly for molecular dynamics and related ones that requires a precise force prediction. That being said, we conduct experiments where the loss weight is adjusted to 0.9*force MAE + 0.1*energy MAE (different to the number reported in Tab. 5). The checkpoint in 2500 epochs shows that it performs SOTA on energy prediction for aspirin in rMD17, as shown in Tab. 10.

**Force field on large molecules.** MD22 is a large molecules benchmark adopted by several studies (Wang et al., 2024; 2023c; Li et al., 2023). As shown in Tab. 3, it reveals that FreeCG also performs well for large molecules. It leads in most tracks for force prediction, and shows comparable results for energy prediction. Remarkably, The decreasing in MAE for energy and force prediction on

Ac-Ala3-NHMe are both around 20%. The performances for the other models are not consistently well for force prediction, while ViSNet-LSRM exhibits strong performance for energy prediction. It is also reasonable that all modern deep neural network-based methods outperform sGDML, as a classical kernel method.

## 4.2 Comparison with state-of-the-arts for molecular properties predictions

To examine the generalization power on molecular property prediction of FreeCG, we collect QM9 as a standard benchmark for this task. FreeCG performs the best for most properties. Although FreeCG is proposed to be MLFFs, but it is even more comparable than others in molecular property prediction tasks. The results in Tab. 6 demonstrate strong generalization capabilities of FreeCG.

## 4.3 Efficiency benchmarking

Except for the conformation exploration in Sec. A.7, AIMD-Chig dataset Wang et al. (2023a) is taken for testing the memory usage and inference speed. It contains mini-protein which is more challenging for inference speed and memory usage. We compare the inference speed and memory usage of FreeCG with ViSNet, NequIP, and Allegro. The results are shown in Fig. 5. The training time and the numbers of parameters are also shown in Tab. 12. FreeCG adds little extra time and memory cost, compared to the baseline model, ViSNet. It is also the most efficient one for both memory and speed, compared to the other two CG transform-based methods, NequIP and Allegro. The overall results prove the effciency of FreeCG. The number of groups in group CG transform also impacts the inference speed. Fig. 3 shows the number of paths and the actual inference time for different group numbers. A computation analysis for CG transform can be referred to Sec. A.6.

## 4.4 Ablation study

We conduct ablations on different modules we propose, as well as the strategies for abstract edges shuffling. The results are shown in Tab. 7. It reveals that each of our module contributes to the final score of FreeCG. In the final implementation of abstract edges shuffling, we add the index of each abstract edge by $\lfloor 1.5 * T/G \rfloor$. Here we also study the influence of the shuffling strategies. We adopt $\lfloor 0.5 * T/G \rfloor$, $\lfloor 1.0 * T/G \rfloor$, and $\lfloor 1.5 * T/G \rfloor$ for comparing the performance. We can see from the result that $\lfloor 1.5 * T/G \rfloor$ works the best. The group numbers are also evaluated and a large number of groups appears to be a good choice. The ablation on sparse path module is reported in Tab. 8. It reveals that sparse path exhibits a satisfying trade-off between performance and efficiency. Full path only decreases minor energy and force prediction MAE, while brings huge overhead in training, inference speed (see Fig. 3), and memory cost.

## 4.5 Extension to other models

FreeCG also presents a paradigm to enhance other geometric models. We use the former SOTA model, QuinNet, as an example to illustrate how effectively FreeCG can be extended to other architectures. QuinNet has a transformer architecture inside, so we construct abstract edges the same way FreeCG does, and adopts all FreeCG modules upon those abstract edges. We evaluate the training curve of 1000 epochs training for both QuinNet+FreeCG and vanilla QuinNet. As shown in Fig. 10 and 11, when equipped with FreeCG, QuinNet significantly gets improved for both energy and force prediction. This trend becomes more pronounced with longer training periods, as evidenced by the results at the 1000*th* epoch. We also report the results of training for 1500 epochs (see Fig. 11), which shows that QuinNet+FreeCG significantly outperforms fully-trained QuinNet.

## 5 Conclusion

This work proposes FreeCG, an equivariant neural network that frees the design space of CG transform. It achieves SOTA performance in force prediction for MD17, rMD17, and MD22 datasets, as well as in molecular properties prediction for QM9, with only minor computational overhead. The practicality of FreeCG for conducting molecular dynamics simulations is thoroughly examined across periodic systems, small molecules in MD17, and the mini-protein Chignolin. As we show that FreeCG helps improve QuinNet, it also introduces a new paradigm for expressive and efficient CG transform-based neural network design in the future.

## 6 ETHICS STATEMENT

It is crucial to ensure that the use of MLFFs is strictly regulated and controlled to prevent their misuse for illegal purposes, such as the development and deployment of chemical weapons. Furthermore, global cooperation and information sharing between the academic community and industry are key to identifying and mitigating potential threats in a timely manner. We require that further research published on top of this work could adhere to the code of ethics, avoiding the misuse of MLFFs. Meanwhile, the data used in our work does not contain any human subjects. All the data we used are freely available on the internet.

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

# A APPENDIX

## A.1 EXPERIMENTAL SETTINGS

We conduct all the experiments under the same software and hardware settings. The machine is equipped with an Intel® Xeon® Gold 6330 CPU @ 2.00GHz, with NVIDIA Tesla A100 80G GPUs. We run the experiments for each molecule on a single GPU. Pytorch 1.10.0 is used as the basic machine learning python library. For the CG transform operations, we adopt e3nn 0.5.1. Matplotlib 3.0.3 is utilized for plotting. The details can be referred to Tab. 4. We report the hyperparameters used in Tab. 5. For training/validation/test splits, we follow previous works (Wang et al., 2024; 2023c; Fu et al., 2022). We pick up the model for evaluating on test set based on the performance on the validation set. If the model does not improve for a given number of epochs, we will terminate the training and select the checkpoint with the best validation score. As previous works, Exponential Moving Average (EMA) is adopted to generate the model weights. The detailed training configurations are shown in Tab. 5.

Table 4: Hardware and software settings.

| Hardware | | Software | | |
|---|---|---|---|---|
| CPU | GPU | Neural Network | Equivariance | Plotting |
| Intel® Xeon® Gold 6330 CPU @ 2.00GHz | NVIDIA Tesla A100 | Pytorch 1.10.0 | e3nn 0.5.1 | Matplotlib 3.0.3 |

Table 5: Hyperparameters for each dataset.

| Hyperparameter | MD17 | rMD17 | MD22 | QM9 | Water-1k | LiPS | Chignolin | OC20 |
|---|---|---|---|---|---|---|---|---|
| Initial learning rate | 4e-4, 2e-4 | 2e-4 | 2e-4, 1e-4 | 1e-4 | 5e-4 | 1e-3 | 2e-4 | 1e-4 |
| Learning rate decay factor | | | | 0.8 | | | | |
| Learning rate decay patience | 30 | 30 | 30 | 15 | 5 | 5 | 10 | - |
| Learning rate warmup step | 1000 | 1000 | 1000 | 10000 | 1000 | 1000 | 1000 | 6000 |
| Loss | | | | Mean Squared Error (MSE) loss | | | | |
| Optimizer | | | | AdamW ($\beta(0.9, 0.999)$) | | | | |
| Epoch | 3000 | 3000 | 3000 | 1500 | 1000 | 100 | 3000 | 12 |
| Batch size | 4 | 4 | 4 | 32 | 1 | 1 | 4 | 4 |
| Number of layers | 9 | 9 | 9 | 9 | 9 | 9 | 6 | 9 |
| Cutoff | 5.0, 4.0 | 5.0 | 5.0, 4.0 | 5.0 | 6.0 | 6.0 | 5.0 | 14 |
| Force/Energy loss weights | 0.95/0.05 | 0.95/0.05 | 0.95/0.05 | - | 1.0/0 | 1.0/0 | 0.95/0.05 | 0.99/0.1 |
| Dimension of latent feature | 256 | 256 | 256 | 512 | 256 | 256 | 128 | 256 |
| Number of groups | | | | 32 | | | | |
| Output head | | | | Equivariant/Scalar | | | | |
| EMA rate | | | | 0.999 | | | | |

## A.2 MODEL IMPLEMENTATION

Here we show how FreeCG is built upon ViSNet. This section provides detailed explanations of the implementation details, ensuring FreeCG can be replicated effectively.

**Input layer.** Given the atom coordinates and types $\{\boldsymbol{X} = \vec{r}_0, \vec{r}_1, \vec{r}_2, ..., \vec{r}_N), \boldsymbol{Z} = (z_1, z_2, ..., z_n)\}$, where $\vec{r} \in \mathbb{R}^3$ the Cartesian coordinates of atom, and $z$ the atom type (atom numbers). First we embed the atom types to the latent space, and take them as our first layer's node features $h_i = \text{embedding}(z_i) \in \mathbb{R}^C$. $C$ is the dimension of the latent space. For each atom, we only consider neighboring atoms within a given radius $\mathcal{N}(i)$, where we maintain the distance vector from the central atom to the neighboring atoms, and *lift* them to $(l = 1, p = -1)$ and $(l = 2, p = 1)$ irreps $E_{ij} \in \mathbb{R}^{3+5}$ via real spherical harmonics applied on the unit vector $E_{ij} = Y^{l=1}(\vec{r}_{ij}/\|\vec{r}_{ij}\|) \oplus Y^{l=2}(\vec{r}_{ij}/\|\vec{r}_{ij}\|)$, where we also calculate the corresponding Euclidean norm $\|\vec{r}_{ij}\|$. The Euclidean norms of vectors are then converted to high-dimension scalar features (edge attributes) $f_{ij} = \text{RBF}(\vec{r}_{ij}) \in \mathbb{R}^C$ by radial basis functions (RBFs). We also maintain zero-initialized abstract edges $\overline{E}_i^{L=0} = \boldsymbol{0}$ for each node to be updated in the following layers. We assign

Table 6: Molecular property prediction on QM9 dataset. The results are reported in MAE. The best numbers are marked in **bold**.

| Target | | SchNet | EGNN | DimeNet++ | PaiNN | SphereNet | PaxNet | ET | ComENet | ViSNet | EquiformerV2 | FreeCG |
|---|---|---|---|---|---|---|---|---|---|---|---|---|
| $\mu$ | mD | 33 | 29 | 29.7 | 12 | 24.5 | 10.8 | 11 | 24.5 | **9.5** | 10 | 11.4 |
| $\alpha$ | $ma_0^3$ | 235 | 71 | 43.5 | 45 | 44.9 | 44.7 | 59 | 45.2 | 41.1 | 50 | **38.2** |
| $\epsilon_{HOMO}$ | meV | 41 | 29 | 24.6 | 27.6 | 22.8 | 22.8 | 20.3 | 23.1 | 17.3 | **14** | 16.6 |
| $\epsilon_{LUMO}$ | meV | 34 | 25 | 19.5 | 20.4 | 18.9 | 19.2 | 17.5 | 19.8 | 14.8 | **13** | 13.5 |
| $\Delta\epsilon$ | meV | 63 | 48 | 32.6 | 45.7 | 31.1 | 31 | 36.1 | 32.4 | 31.7 | **29** | 31.5 |
| $\langle R^2 \rangle$ | $ma_0^2$ | 73 | 106 | 331 | 66 | 268 | 93 | 33 | 259 | **29.8** | 186 | 82.1 |
| $ZPVE$ | meV | 1.7 | 1.55 | 1.21 | 1.28 | 1.12 | 1.17 | 1.84 | 1.2 | 1.56 | 1.47 | **1.10** |
| $U_0$ | meV | 14 | 11 | 6.32 | 5.85 | 6.26 | 5.9 | 6.15 | 6.59 | 4.23 | 6.17 | **4.11** |
| $U$ | meV | 19 | 12 | 6.28 | 5.83 | 6.36 | 5.92 | 6.38 | 6.82 | **4.25** | 6.49 | 4.51 |
| $H$ | meV | 14 | 12 | 6.53 | 5.98 | 6.33 | 6.04 | 6.16 | 6.86 | 4.52 | 6.22 | **4.13** |
| $G$ | meV | 14 | 12 | 7.56 | 7.35 | 7.78 | 7.14 | 7.62 | 7.98 | 5.86 | 7.57 | **5.65** |
| $C_v$ | mcal/mol K | 33 | 31 | 23 | 24 | 22 | 23.1 | 26 | 24 | 23 | 23 | **20.4** |

Table 7: Ablation on different modules. Abstract edges shuffling and Attention enhancer are added upon the best choices of the above modules, with respect to the validation loss.

| Method | Aspirin | | |
|---|---|---|---|
| | Val loss | Energy | Force |
| ViSNet | - | 0.116 | 0.155 |
| + Group CG transform | | | |
| 8 groups | 0.0509 | 0.123 | 0.144 |
| 32 groups | 0.0416 | 0.112 | 0.129 |
| + Abstract edges shuffling | | | |
| 1-group shuffle | 0.0401 | 0.112 | 0.128 |
| 0.5-group shuffle | 0.0396 | 0.110 | 0.128 |
| 1.5-group shuffle | 0.0384 | 0.111 | 0.125 |
| + Attention enhancer | **0.0345** | **0.110** | **0.122** |

the same number of abstract edges as the dimension of the latent features, such that additional operations to align the dimension numbers are not required.

**Intermediate layers.** Here, we use a superscript $L$ to denote the index of layer that the features are in. The message-passing between atoms is implemented by a transformer architecture. For each atom $i$, the neighboring atoms $j \in \mathcal{N}(i)$ will send messages to $i$, and the messages are aggregated to update the information of $i$. The query, key, and value of the node features are first calculated, respectively: $q_i = f_q(h_i)$, $k_j = f_k(h_j)$, $v_j = f_v(h_j)$. The edge attributes are also converted to auxiliary terms $dk_j = f_{dk}(f_{ij})$ and $dv_j = f_{dv}(f_{ij})$ to modulate keys and and values of atoms. Here functions $f$'s are all fully-connected linear operations. Then we calculate the cross-attention between $i$ and $j$, which is

$$a_{ij} = \text{SiLU}\left( \text{Cutoff}(\|\vec{r}_{ij}\|) q_i k_j dk_j + \text{AttEnhancer}(E_{ij}, \overline{E}_j^L) \right) \qquad (7)$$

where $\text{Cutoff}(\cdot)$ is a cosine cutoff function, and $\text{AttEnhancer}(\cdot)$ the proposed attention enhancer module, as we will formulate its details. First, recall the dimension of $\overline{E}_i^L \in \mathbb{R}^{C*8}$ and $E_{ij} \in \mathbb{R}^8$. Each of the $C$ abstract edges will undergo a dot product with $E_{ij}$. The highest value among them will be the output of AttEnhancer. In other word,

$$\text{AttEnhancer}(\overline{E}_i^L, E_{ij}) = \max_C(\overline{E}_i^L \odot E_{ij}) \qquad (8)$$

as we introduce in Eq. 6. Then the values are multiplied with $dv$ and attention.

$$\hat{v}_{j \mapsto i}^L = v_j \cdot dv_j \cdot a_{ij} \qquad (9)$$

It then undergoes two different fully-connected operations to generate two coefficients $s_1$ and $s_2$. They are used to generate the abstract edges:

$$\hat{E}_{j \mapsto i}^L = \overline{E}_i^L \cdot s_1 + E_{ij} \cdot s_2 \qquad (10)$$

This variable, together with $\hat{v}_{j \mapsto i}$, are aggregated by sum:

$$\hat{E}_i^L = \sum_{j \in \mathcal{N}(i)} \hat{E}_{j \mapsto i}^L \qquad (11)$$

Table 8: Ablation on Sparse Path. The energy and force prediction errors are reported in mean abosolute error (MAE). The unit of training speed is iterations/second. The mini-batch size is set to 32.

| | Force MAE | Energy MAE | Train. speed | Num. of param. |
|---|---|---|---|---|
| Sparse path | 0.122 | 0.110 | 3.62 | 10.4M |
| Full path | 0.118 | 0.106 | 0.79 | 43.8M |

$$\hat{v}_i^L = \sum_{j \in \mathcal{N}(i)} \hat{v}_{j \mapsto i}^L \tag{12}$$

$\hat{v}_i^L$ then converts to three variables for further operation:

$$o_i^{L,1}, o_i^{L,2}, o_i^{L,3} = \text{Linear}(\hat{v}_i^L) \tag{13}$$

$\hat{E}_i^L \in \mathbb{R}^{C*8}$ and $\overline{E}_i^L \in \mathbb{R}^{C*8}$ are used for the following group CG transform and abstract edges shuffling. First $\overline{E}_i^L \in \mathbb{R}^{C*8}$ undergoes a fully-connected operation along $C$ dimension, and multiply with $o_i^{L,1}$, which means we get $o_i^{L,1} \cdot \text{Linear}(\overline{E}_i^L)$.

It, together with $\hat{E}_i^L \in \mathbb{R}^{C*8}$, are then divided into $G$ groups along $C$ dimension, where we get $\hat{E}_{i,t \in G_g}^L \in \mathbb{R}^{\frac{C}{G}*8}$, and $(o_i^{L,1} \cdot \text{Linear}(\overline{E}_i^L))_{t \in G_g} \in \mathbb{R}^{\frac{C}{G}*8}$. Then, we perform CG transform between two variables in a fully connected form with learnable weights, and concatenate the results to generate $d\overline{E}_i'^{L+1}$ before shuffling, as shown in Eq. 5. For the shuffling strategies, we add $\lfloor \frac{3C}{2G} \rfloor$ to each index of the abstract edges $\overline{E}_i'^{L+1}$. Then, it is added with $\hat{E}_i^L$ to form a residual structure, as we show here:

$$d\overline{E}_i^{L+1} = \text{shuffle}(d\overline{E}_i'^{L+1}) + \hat{E}_i^L \tag{14}$$

where $d\overline{E}_i^{L+1}$ is added to $\overline{E}_i^L$ to obtain $\overline{E}_i^{L+1}$. Next, we update $h$ and $f$. We first show the update for $h$:

$$dh_i^{L+1} = h_i^L + \left( \text{Linear}_1(\overline{E}_i^L) \odot \text{Linear}_2(\overline{E}_i^L) \right) \cdot o_i^{L,2} + o_i^{L,3} \tag{15}$$

To update $f$, we follow ViSNet to leverage rejection of vectors:

$$df_{ij}^{L+1} = f_{ij}^L + \text{RejCalc}_{\text{trg}}(\overline{E}_i^L, \vec{r}_{ij}) \odot \tag{16}$$
$$\text{RejCalc}_{\text{src}}(\overline{E}_i^L, \vec{r}_{ij}) \cdot \text{SiLU}(\text{Linear}(f_{ij}^L))$$

where rejection calculation module $RejCalc$ is:

$$RejCalc_{mode}(a, b) = a - (\text{Linear}_{\text{mode}}(a) \odot b) \cdot b \tag{17}$$

The updated $\overline{E}^{L+1}$, $h^{L+1}$, and $f^{L+1}$ are fed into the next layer.

**Output layers** are different with respect to the task our model performs on. We introduce the details for each task.

*Force field prediction.* FreeCG is based on energy-conservative field, which means we derive the force from the predicted potential energy. Following ViSNet (Wang et al., 2024) and PaiNN (Schütt et al., 2021), we predict the potential energy of the molecule via equivariant gated module.

$$h_i^{L+1}, u_i^{L+1} = \text{MLP}\left( \text{Concat}(h_i^L, \|\text{Linear}_1(\overline{E}_i^L)\|) \right) \tag{18}$$

where MLP is an 1-hidden layer multi-layer preceptor. There is one more step to update $\overline{E}_i^{L+1}$:

$$\overline{E}_i^{L+1} = \text{Linear}_2(\overline{E}_i^L) \cdot u_i^{L+1} \tag{19}$$

These calculations are then repeated twice in succession. There is also an alternative head design for force field prediction which uses only scalar features $h$. Under this setting, Eq. 18 and (19) are replaced by ($\boldsymbol{L}$ denotes the last layer):

$$h^{\boldsymbol{L}} = \text{MLP}(h^{\boldsymbol{L}-1}) \tag{20}$$

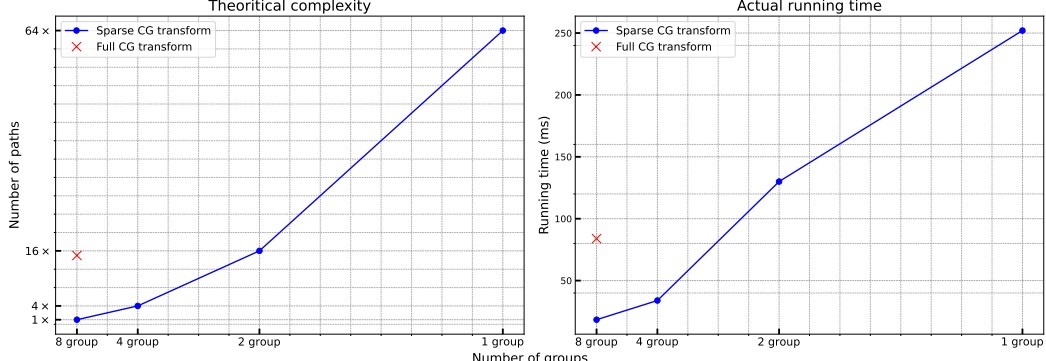

Figure 3: Efficiency analysis of group CG transform. **Left:** The number of paths for CG transform under different group numbers, where the numbers of irreps are the same. **Right:** The actual running time for CG transform for different group numers. Here we adopt sparse path strategy for computing 512 irreps (before grouping) for each $l$. Full CG transform denotes not using sparse path.

In Tab. 5 where we introduce our hyper-parameter choices, the scalar version of output head is denoted as *Scalar*, and the other one *Equivariant*. Finally, the total energy of the molecule is the sum of the last-layer node features $h_i^{\boldsymbol{L}} \in \mathbb{R}$:

$$y = \sum_i h_i^{\boldsymbol{L}} \tag{21}$$

and the force is the negative gradients of the total energy:

$$F_i = -\nabla_{\vec{r}_i} y \tag{22}$$

*Property prediction.* The calculations for properties in QM9 follow the same procedure as energy prediction in force field prediction, with the exception of molecular dipole and electronic spatial extent. We first need to calculate the center of mass $\vec{r}_c$, which is:

$$\vec{r}_c = \frac{\sum_i m_i \cdot \vec{r}_i}{\sum_i m_i} \tag{23}$$

For molecular dipole, the formula is:

$$\mu = \left\| \sum_i \overline{E}_i^{\boldsymbol{L}} + h_i^{\boldsymbol{L}}(\vec{r}_i - \vec{r}_c) \right\| \tag{24}$$

and for electronic spatial extent:

$$\langle R^2 \rangle = \sum_i h_i^{\boldsymbol{L}} \|\vec{r}_i - \vec{r}_c\| \tag{25}$$

The output head can be easily adapted for different tasks, providing flexibility in property prediction.

### A.3 DATASETS DETAILS

**MD17 and rMD17.** They are both molecular dynamics datasets for small molecules. MD17 (Chmiela et al., 2017), proposed by Chmiela, S., et al. contains *ab-initio* level molecular dynamics trajectories. Four types of data are included in the dataset: atomic numbers, atomic positions, molecular energy, and the force acting on each atom. To alleivate the noise during the trajectory computation, Christensen, A. S. et al. also propose revised MD17 (rMD17) (Christensen & Von Lilienfeld, 2020), where molecular trajectories are calculated at the PBE/def2-SVP level of theory. The precision of the calculated trajectories is upheld by the tight SCF convergence and dense DFT integration grid.

**MD22** consists of larger molecules with atoms numbering from 42 to 370, in contrast to MD17 and rMD17. The trajectories are sampled between 400K and 500K at 1fs resolution. The energy and

Table 9: The basic operation number for each type of CG transform. $l_o$ denotes the output degree. The column and row numbers represent the degrees of two input irreps, respectively. The cyan blocks represent the operations in regular neural networks, while the others are for high-order CG transform.

| $l_o = 2$ | 0 | 1 | 2 | $l_o = 1$ | 0 | 1 | 2 | $l_o = 2$ | 0 | 1 | 2 |
|---|---|---|---|---|---|---|---|---|---|---|---|
| 0 | 1 | | | 0 | - | 3 | - | 0 | - | - | 5 |
| 1 | | 3 | | 1 | 3 | 6 | 9 | 1 | - | 9 | 12 |
| 2 | | | 5 | 2 | - | 9 | 12 | 2 | 5 | 12 | 19 |

Table 10: Performances on MD17 dataset under loss weight being 0.9*force MAE+0.1*energy MAE. The results are reported in mean absolute error (MAE). The energies and forces are measured in kcal/mol and kcal/mol/Å, respectively. The best numbers are marked in **bold**.

| | | UNiTE | GemNet | NequIP | MACE | Allergo | BOTNet | VisNet | QuinNet | ACE$^g$ | PϴNITA | FreeCG |
|---|---|---|---|---|---|---|---|---|---|---|---|---|
| Aspirin | Energy | 0.055 | - | 0.0530 | 0.0507 | 0.0530 | 0.0530 | 0.0445 | 0.0486 | 0.0392 | 0.0392 | **0.0374** |
| | Force | 0.175 | 0.2191 | 0.1891 | 0.1522 | 0.1684 | 0.1900 | 0.1520 | 0.1429 | 0.1407 | 0.1338 | **0.1225** |
| Azobenzene | Energy | 0.025 | - | 0.0161 | 0.0277 | 0.0277 | 0.0161 | 0.0156 | 0.0394 | **0.0115** | 0.0161 | 0.0181 |
| | Force | 0.097 | - | 0.0669 | 0.0692 | 0.0600 | 0.0761 | 0.0585 | 0.0513 | 0.0507 | 0.0530 | **0.0475** |
| Benzene | Energy | 0.002 | - | 0.0009 | 0.0092 | 0.0069 | 0.0007 | 0.0007 | 0.0096 | **0.0002** | 0.0039 | 0.0084 |
| | Force | 0.017 | 0.0115 | 0.0069 | 0.0069 | 0.0046 | 0.0069 | 0.0056 | 0.0047 | **0.0023** | 0.0069 | 0.0057 |
| Ethanol | Energy | 0.014 | - | 0.0092 | **0.0032** | 0.0092 | 0.0092 | 0.0078 | 0.0096 | 0.0046 | 0.0592 | 0.0072 |
| | Force | 0.085 | 0.083 | 0.0646 | 0.0484 | 0.0484 | 0.0738 | 0.0522 | 0.0516 | **0.0322** | 0.0577 | 0.0450 |

force labels are obtained at the PBE+MBD level of theory. The root mean squared test error of force prediction is controlled to be around 1 kcal/mol/Å in the original paper (Chmiela et al., 2023). Thus, the training data sizes for different molecules vary. Generally, the larger the molecules, the smaller the training data size.

**QM9** consists of around 130,000 molecules with 12 properties regression tasks. It is a subset of the GDB-17 database (Ruddigkeit et al., 2012). The data is calculated at B3LYP/6-31G(2df,p) based DFT level of accuracy. Since the attributes vary for different properties, we use distinct output head for each, as discussed in Sec. A.2.

**Chignolin.** The AIMD-Chig dataset (Wang et al., 2023a) comprises of two million conformations of the 166-atom protein chignolin, obtained through sampling at the M06-2X/6-31 G* based DFT level. There are approximately 10,000 different conformations, including folded, unfolded, and metastable states. We report the performances of FreeCG on different parts of the energy landscape, and adopt this dataset to benchmark efficiency, following (Wang et al., 2024).

**Periodic systems.** Periodic Boundary Conditions (PBCs) are vital in molecular dynamics simulations of periodic systems as they eliminate surface effects, enhance statistical sampling, and provide a realistic representation of bulk properties. Here we focus on two typical molecules, water and LiPS. The water dataset are generated by the flexible version of the Extended Simple Point Charge water model (SPC/E-fw) (Wu et al., 2006) in (Fu et al., 2022). The authors provides with several sizes of training sets, including -1k, -10k, 90k. Here we adopt the 1k version, where 950 samples are for training and the rest for validation. LiPS is an important solid-state materials for battery development. It can help predict key performance metrics such as capacity, energy density, and cycle life, aiding in the development of next-generation lithium-ion batteries. We follow the same train/val split as (Fu et al., 2022). Some 3D structures of the data can be referred to Fig. 7.

A.4 PROOF OF THE PERMUTATION INVARIANCE OF ABSTRACT EDGES

According to Sec. A.2, we first recall the last step for generating abstract edges:

$$\hat{E}_i^L = \sum_{j \in \mathcal{N}(i)} \hat{E}_{j \mapsto i}^L = \sum_{p \in P} \sum_{j \in \mathcal{N}(i)} \frac{\mathcal{P}(p)\hat{E}_{j \mapsto i}^L}{\text{Card}(P)} \tag{26}$$

Figure 4: Applications for the 166-atom mini-protein, Chignolin. **a.** The energy landscape of Chignolin was sampled using Replica Exchange Molecular Dynamics (REMD). This landscape is characterized by two key distance parameters: the x-axis represents the distance between the carbonyl oxygen on the D3 backbone and the nitrogen on the G7 backbone, while the y-axis depicts the distance between the carbonyl oxygen on the E5 backbone and the nitrogen on the T8 backbone. These two distance metrics collectively illustrate the conformational states of Chignolin across its energy landscape. The left and right energy basins are corresponded to folded and unfolded states, respectively. **b.** The six conformations are sampled at the localization highlighted in the energy landscape. The force and energy performances (kcal/mol) are reported, with a comparison made to ViSNet. These six conformations cover both folded and unfolded states. **c.** The RMSD (Å) during the molecular dynamics simulation. The shaded area denotes the values of standard derivations. The RMSD values are obtained by taking average of 10 trajectories. Better viewed in color.

where $P$ is the set for all permutation operations, here we omit the subscript of $\mathcal{P}$ for specific spaces to work on. This equation indicates that, performing our aggregation (the middle part) is equivalent to performing the last part (which guarantees the operation is permutation invariant). Here, note we are proving that the abstract edges for each atom are permutation invariant, and we can freely design CG transform *per atom*, thus the permutation is applied to $j$ but not $i$. It sums over all the permutation operations, and thus *the last step* is permutation invariant. Then, it suffices to show that each of the previous step are all at least permutation equivariant. It also suffices to show they are permutation equivariant *w.r.t.* single index switch operation, as each permutation operation can be made by several switches. If we exchange, without loss of generality, index $x$ and $y$, then those $a_{ij}$ that $x$ or $y$ shows up in the subscript for $j$ will exchange with each other, and so do $\hat{v}_{j\mapsto i}$ and $\hat{E}^L_{j\mapsto i}$. Thus, the rest steps are equivariant *w.r.t.* single switch, and so they are permutation equivariant. Therefore, we conclude our proof that abstract edges $\hat{E}$'s are permutation invariant. For $\overline{E}$, it is also permutation invariant, as it is generated by permutation invariant function $h$ in the previous layer. $\overline{E}^{L=1}$ in the first layer is permutation invariant too, as it is set to be fixed zeros.

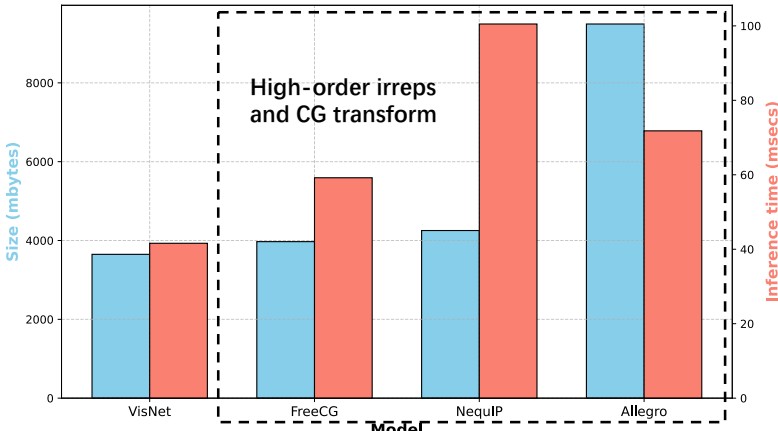

Figure 5: The speed and memory occupation of FreeCG compared with other SoTA models. Numbers are reported based on a single chignolin molecule. The right three models are based on high-order irreps and CG transform. Better viewed in color.

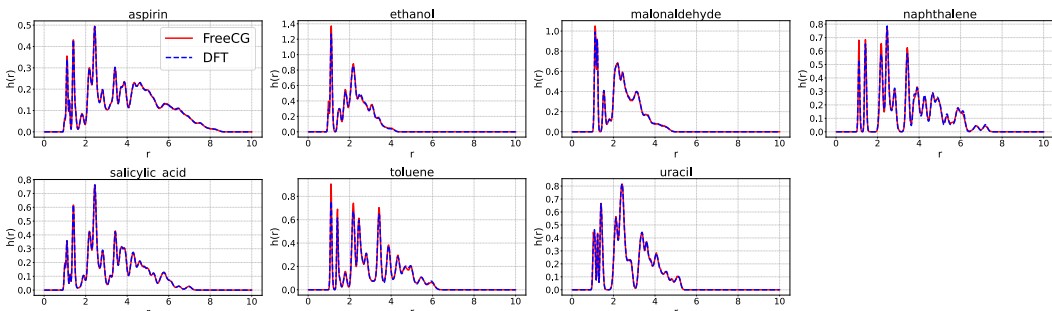

Figure 6: The distributions of interatomic distances $r$ during molecular dynamic simulations of MD17. The unit of $r$ is Å, and the unit of $h(r)$ is Å$^{-1}$.

### A.5 PROOF OF THE EQUIVARIANCE OF THE LINEAR COMBINATION OF SPHERICAL IRREDUCIBLE REPRESENTATION

Let $v_i$ be the spherical irreps with order $l$, $g \in SO(3)$, $w_i$ the weight for $i$-th vector in the weighted sum, and $\rho_V$ the group homomorphism. We need to prove $\rho_V(g) \sum_i w_i v_i = \sum_i w_i \rho_{V_i}(g) v_i$. It holds if we can prove the following conditions 1) $\rho_{V_i}(g) w_i v_i = w_i \rho_{V_i}(g) v_i$ which is trivially true as $w_i$ is a scalar; 2) $\rho_{V_i} = \rho_{V_j}$ for arbitrary $i$ and $j$.

To show 2) is golden, we need to show a) $V_i$ is an invariant subspace of SO(3), since the $\rho_{V_i}$ would be ill-defined if it is not the case, which is also trivially true because that is how spherical spaces are generated; b) $V_i = V_j$ for arbitrary $i$ and $j$. Here, $v_i$ are spherical irreps with same $l$. Spherical space of order $l$ is the same vector space spanned by spherical harmonics $Y_m^l$s. Spherical irreps with order $l$ are the vectors in the same order $l$ spherical space, so $V_i = V_j$ for arbitrary $i$ and $j$. Since all $V_i$s are the same we can simply denote each of them as $V$. Summarize over $i$ for both sides of $\rho_{V_i}(g) w_i v_i = w_i \rho_{V_i}(g) v_i$, we get $\rho_V(g) \sum_i w_i v_i = \sum_i w_i \rho_V(g) v_i$. This concludes our proof.

### A.6 ANALYSIS ON THE EFFICIENCY OF CG TRANSFORM

The CG transform consists of two steps: 1) performing a tensor product between two irreps, and 2) decomposing the resulting tensors into irreps. These transforms are actually quadratic homogeneous polynomials. For the sake of convenience, we discuss SO(3) group here. Recall the CG transform formula:

Table 11: Performances of QuinNet equipped with FreeCG on MD17. The results are reported in MAE. The energies and forces are measured in kcal/mol and kcal/mol/Å, respectively. The best numbers are marked in **bold**.

|  |  | Aspirin | Ethanol | Malonaldehyde | Naphthalene | Salicylic acid | Toluene | Uracil |
|---|---|---|---|---|---|---|---|---|
| QuinNet | Energy | 0.119 | 0.050 | 0.078 | 0.101 | **0.101** | 0.080 | 0.096 |
| (from paper) | Force | 0.145 | 0.060 | 0.097 | 0.039 | 0.080 | 0.039 | 0.062 |
| QuinNet | Energy | 0.132 | 0.052 | 0.076 | 0.109 | 0.106 | 0.081 | 0.099 |
| (1500 epochs) | Force | 0.152 | 0.065 | 0.113 | 0.043 | 0.080 | 0.039 | 0.061 |
| QuinNet+FreeCG | Energy | **0.113** | **0.048** | **0.073** | **0.094** | 0.103 | **0.078** | **0.094** |
| (1500 epochs) | Force | **0.127** | **0.056** | **0.095** | **0.034** | **0.073** | **0.037** | **0.057** |

Table 12: Comparison of parameters and training speed. The mini-batch size is set to 32.

|  | QuinNet+FreeCG | QuinNet | FreeCG | VisNet |
|---|---|---|---|---|
| Number of parameters (M) | 9.4 | 9.1 | 10.4 | 9.8 |
| Iterations per second | 4.19 | 4.45 | 3.62 | 3.95 |

$$D_{m_d}^{l_a l_b \mapsto l_d} = \sum_{m_a, m_b} C_{m_d m_a m_b}^{l_d l_a l_b} A_{m_a}^{l_a} B_{m_b}^{l_b} \tag{27}$$

where $m_a + m_b = m_d$. To illustrate, if we regard single multiplication and addition as the two basic operations, then combining two $l = 1$ irreps to form a $l = 2$ irreps will use up 1 basic operation for $m_d = \pm 2$, 3 operations for $m_d = \pm 1$, and 5 operations for $m_d = 0$, making a total of 13 basic operations. One effective approach to understanding irreps is to view them as an extension of vectors and scalars. The dot product between vectors requires only 5 basic operations, in contrast to the 13 operations mentioned earlier. Hence, the CG transform is extremely time-consuming. The table of basic operations for the CG transform between each pair is shown in Tab. 9.

## A.7 Additional experiments

**Applications for full-atom proteins.** AIMD-Chig dataset (Wang et al., 2023a) comprises nearly 10,000 conformations of the 166-atom mini protein, Chignolin. These conformations were obtained including folded, unfolded, and metastable states. It is important to evaluate the performance of FreeCG on such real-world proteins. Following ViSNet (Wang et al., 2024), as shown in Fig. 4, we explore the energy landscape of Chignolin, where we sample six conformations located at different parts of the landscape, covering folded and unfolded states. The energy and force predictions on these six conformations are compared with ViSNet. FreeCG succeeds in outperforming ViSNet for the most sampled conformations. Molecular dynamics simulations are run from six different initial conformations. To assess the simulation stability, we calculate the Root Mean Square Deviation (RMSD) between each step in the trajectory and the initial conformation, shown in Fig. 4(**c**). The results demonstrate satisfying performance of FreeCG on real-world proteins.

**Applications for molecular dynamics simulation.** We conduct molecular dynamics simulations for FreeCG on MD17 and compare the results with DFT calculations. We run a 300ps simulation for each molecule. The time step is set to 0.5 fs, under a Nosé–Hoover thermostat at 500K temperature. Like previous works (Fu et al., 2022; Wang et al., 2024; 2023c), we are interested in the distribution of interatomic distances. Here, $h(r)$ is defined as the probability density function of interatomic distances $r$. We plot the distribution as $h(r)$, where $h(r)$ are averaged along frames or predicted trajectories, and we desire the distribution is similar to the MD17 datasets calculated by DFT. The results are shown in Fig. 6, which shows that FreeCG is capable to well recover the interatomic distances distribution. We also conduct molecular dynamics simulation on water (1k version) (Fu et al., 2022; Wu et al., 2006) and LiPS (Batzner et al., 2022) datasets under periodic boundary conditions (PBCs) to evaluate how FreeCG performs on large molecular systems. We set the timestep to 0.25 fs and run total 200,000 steps for each type of molecules. We focus on the recovery of radial distribution functions (RDFs) because they effectively describe structural and thermodynamic properties. It is similarly calculated as $h(r)$ but under different constant multipliers. FreeCG finishes all the

Table 13: Force prediction results on OC20 S2EF validation set. The unit of force and energy MAE are eV/Å and eV, respectively. The best numbers are marked in **bold**.

| | Id | OOD Ads | OOD Cat | OOD Both |
|---|---|---|---|---|
| | Force Prediction | | | |
| VisNet | 0.0599 | 0.0673 | 0.0597 | 0.0821 |
| FreeCG | **0.0535** | **0.0614** | **0.0536** | **0.0717** |
| | Energy Prediction | | | |
| VisNet | 0.8807 | 1.0278 | 0.8989 | 1.3175 |
| FreeCG | **0.7157** | **0.7586** | **0.7294** | **0.8938** |

dynamics simulation with accurate recovery of the atomic distributions. The results for molecules under PBCs are shown in Fig. 8 and 9.

**Results on OC20 dataset, S2EF task.** OC20 dataset Chanussot et al. (2021) contains 1.2 million DFT relaxations of molecular adsorption onto catalytic surfaces, serving as a large-scale benchmark for three key tasks in catalysis simulations: Structure to Energy and Force (S2EF), Initial Structure to Relaxed Structure (IS2RS), and Initial Structure to Relaxed Energy (IS2RE). We focus on the S2EF task, which involves predicting energy and forces based on atomic positions and types under Periodic Boundary Conditions (PBCs). Unlike other MLFF datasets, OC20 incorporates both organic and inorganic materials relevant to catalysis research, which is also a critical field for climate and energy applications. The dataset contains variably sized training sets and validation sets with distinct out-of-domain conditions: in-domain (ID), out-of-domain adsorbates (OOD Ads), out-of-domain catalysts (OOD Cat), and combined out-of-domain (OOD Both). Using the OC20 200K training set, we evaluated our model on all validation sets. As shown in Table 13, FreeCG demonstrates significant performance advantages over VisNet across all OC20 benchmarks, highlighting its domain adaptability.

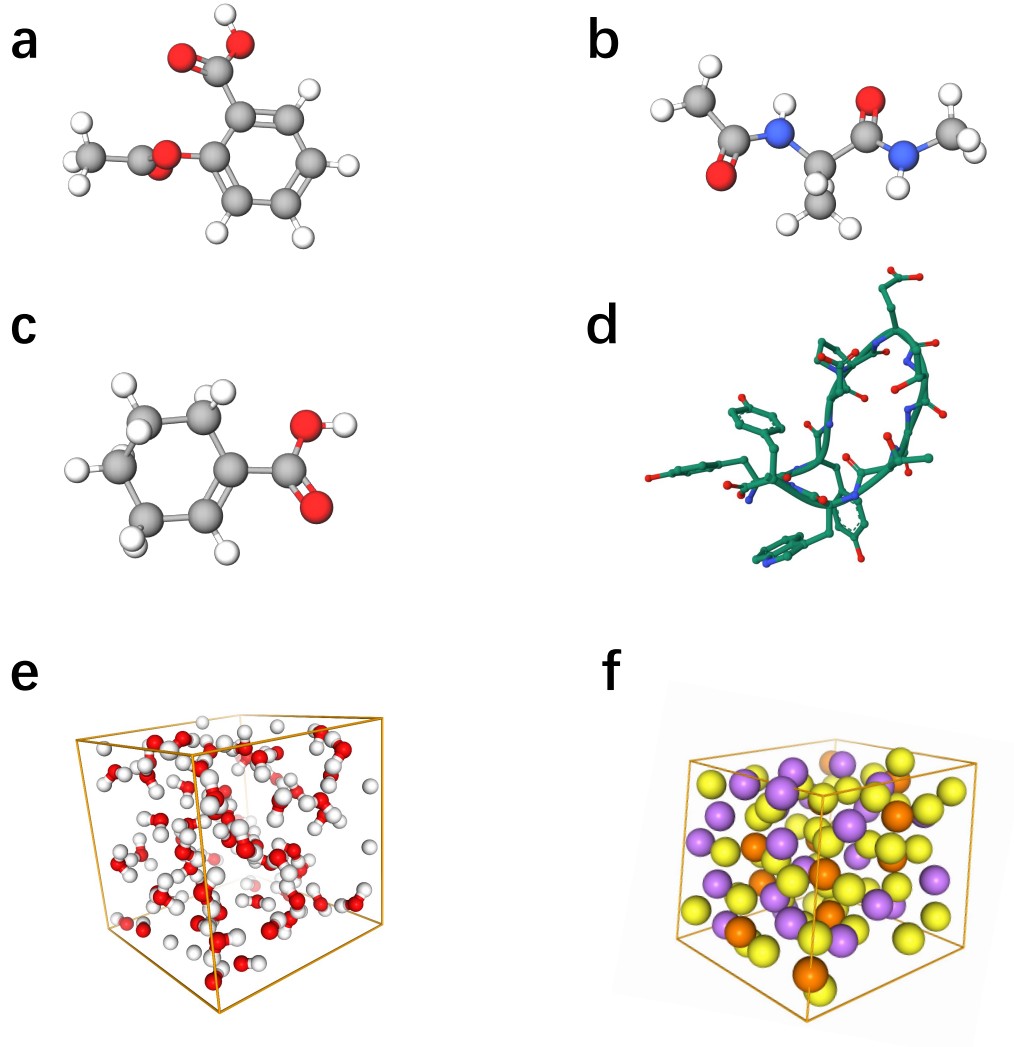

Figure 7: The 3D structures of the data considered in this work. **a.** Aspirin in MD17 and rMD17. **b**. Ac-Ala3-NHMe in MD22. **c.** 1-Cyclohexene-1-carboxylic acid in QM9. **d.** Chignolin. **e.** A single cell of water molecules under PBCs. **f.** A single cell of LiPS under PBCs (Note that the cell is not cubic).

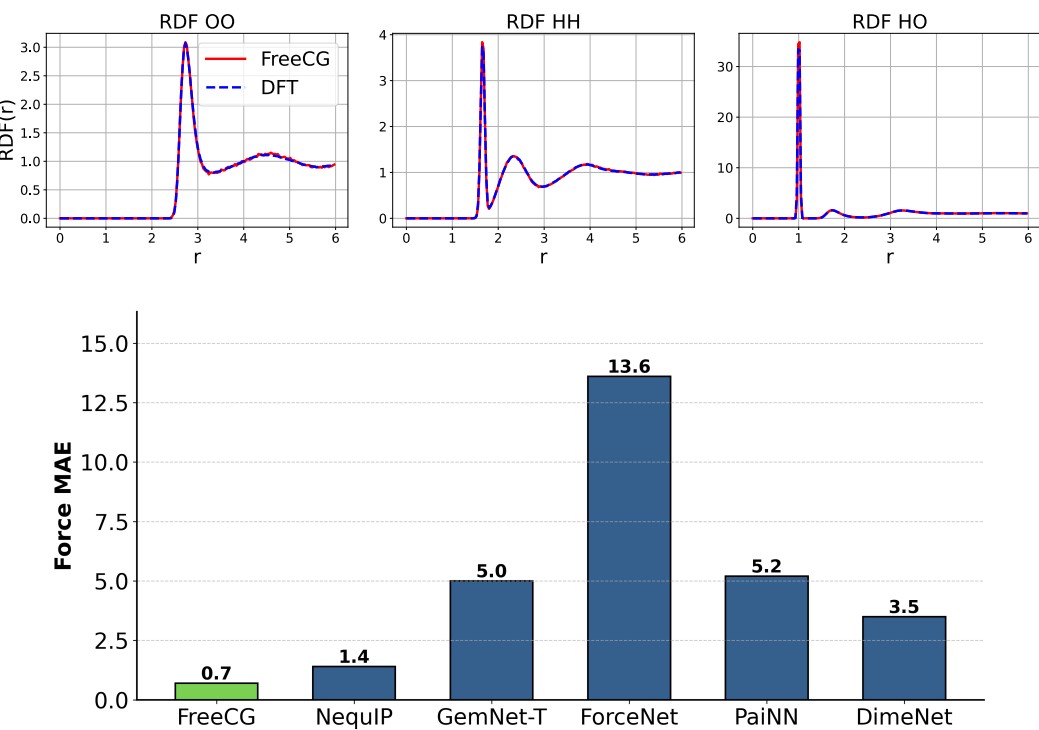

Figure 8: Results on water-1k. **Top:** The RDFs in molecular dynamic simulations for each bond of water under PBCs. The unit of $r$ is Å, and the unit of $\mathrm{RDF}(r)$ is $\mathring{\mathrm{A}}^{-1}$. **Bottom:** The force MAE comparison with other methods. The unit is meV/Å.

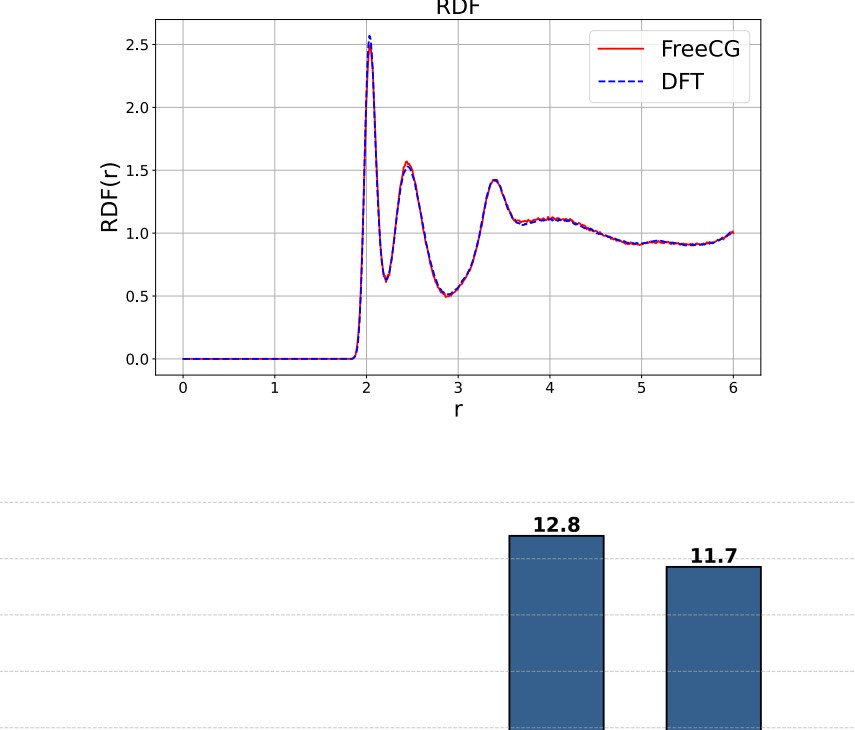

Figure 9: Results on LiPS. **Top:** The RDFs in molecular dynamic simulations for LiPS under PBCs. The unit of $r$ is Å, and the unit of $\mathrm{RDF}(r)$ is $\text{Å}^{-1}$. **Bottom:** The force MAE comparison with other methods. The unit is meV/Å.

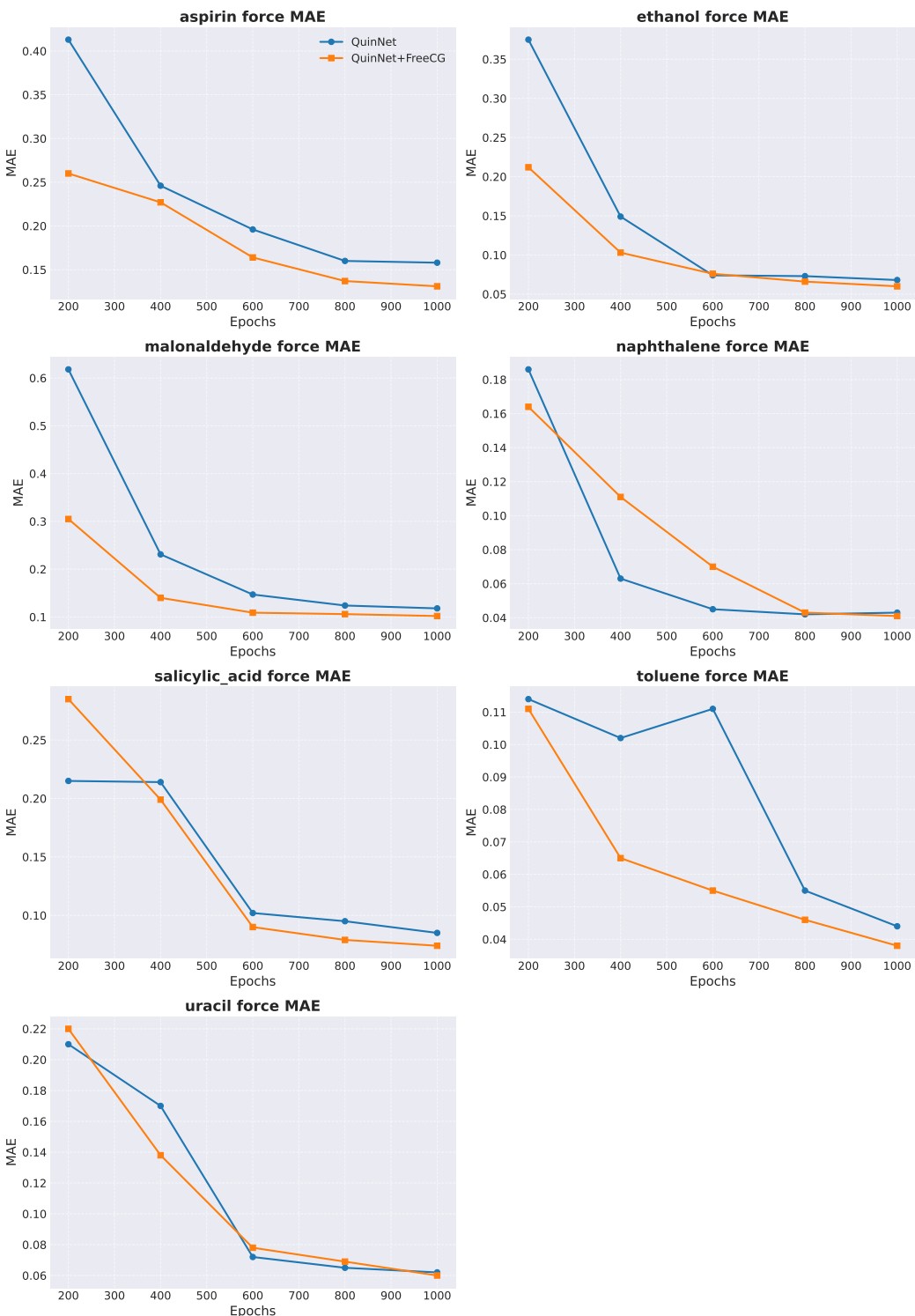

Figure 10: QuinNet force prediction performance on MD17 when equipped with modules from FreeCG. The unit of force is kcal/mol/Å.

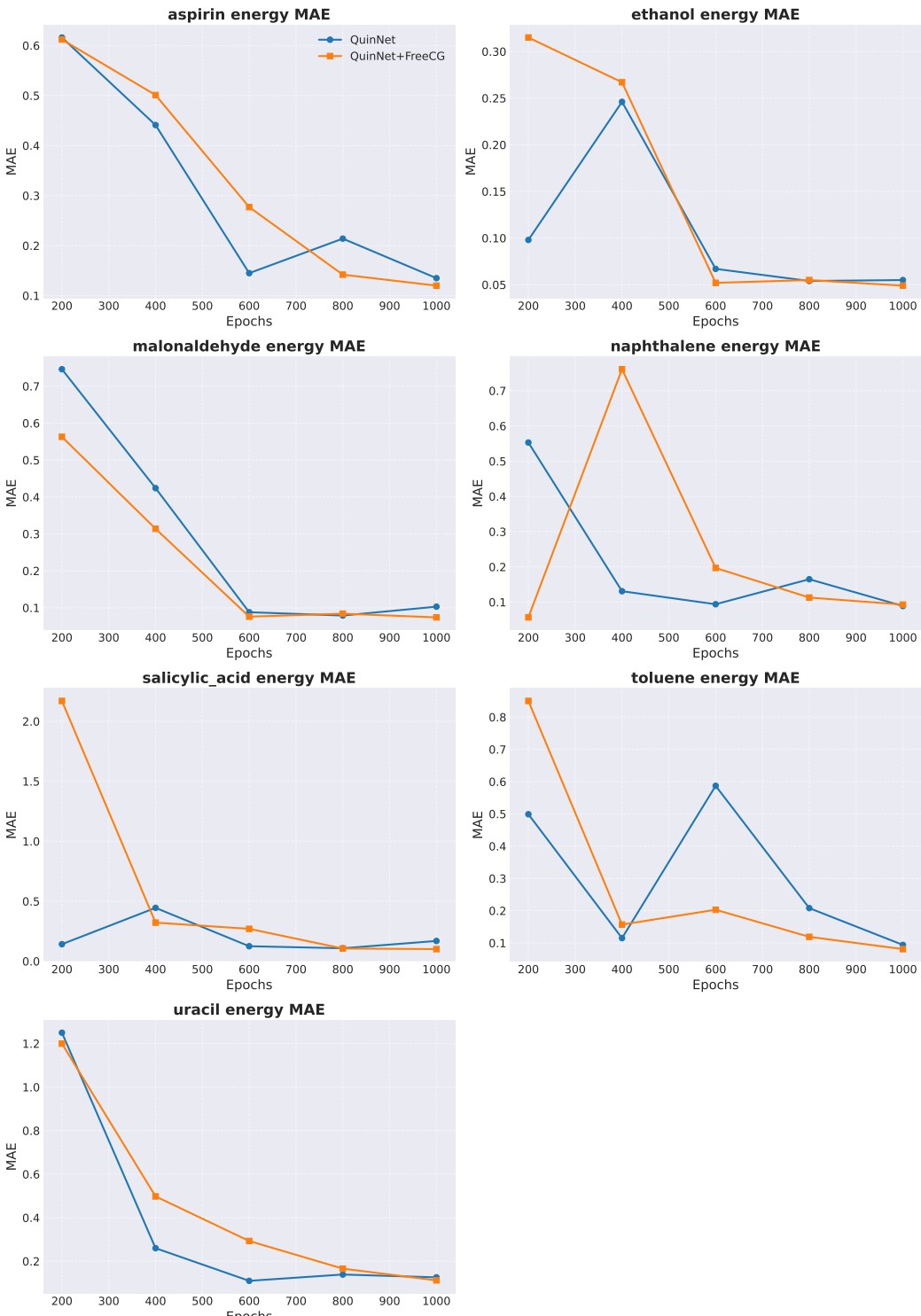

Figure 11: QuinNet energy prediction performance on MD17 when equipped with modules from FreeCG. The unit of energy is kcal/mol.

