# OpenReview forum: "FreeCG: Free the Design Space of Clebsch-Gordan Transform for Machine Learning Force Fields"
_ICLR.cc/2025/Conference — ICLR 2025 Poster_

### Official Review · Reviewer_h6Dk · 2024-10-22

**Soundness:** 3
**Presentation:** 2
**Contribution:** 3
**Rating:** 8
**Confidence:** 5

**Summary:**

The paper introduced an informative and efficient model utilizing high-order irreps and CG transform. This model achieves state-of-the-art (SOTA) results in force prediction for MD17, rMD17, and MD22. However, while the authors asserted that this work represented a new paradigm for CG transform that can be integrated with other models, such as QuinNet, to enhance performance, the experimental results do not support this claim.

**Strengths:**

The paper proposed an informative and efficient model with high-order irreps and CG transform.

The model achieved state-of-the-art (SOTA) results in force prediction for MD17, rMD17, MD22.

**Weaknesses:**

Although the author claimed that this work presented a new paradigm for CG transform that can be combined with other models, such as QuinNet, to achieve better performance, the experimental results did not demonstrate this. Simply analyzing the training curves did not allow readers to determine the impact of adding or removing FreeCG on QuinNet. Furthermore, while other models were trained for 3000 epochs, QuinNet was only trained for 1000 epochs. At this point, there is no clear advantage for QuinNet+FreeCG, and readers cannot ascertain whether extended training would improve QuinNet's performance.

**Questions:**

For MILP, both model execution speed and memory usage are critically important. The proposed method seems to significantly increase the computational complexity of the model. It is crucial to include a comparison of the number of parameters and training speed.

---

> ### Author Response · Authors · 2024-11-14
> **Response to Reviewer h6Dk**
>
> Dear Reviewer,
>
> Thank you for your time in reviewing our paper and providing valuable comments.
>
> > Although the author claimed that this work presented a new paradigm for CG transform that can be combined with other models, such as QuinNet, to achieve better performance, the experimental results did not demonstrate this. Simply analyzing the training curves did not allow readers to determine the impact of adding or removing FreeCG on QuinNet. Furthermore, while other models were trained for 3000 epochs, QuinNet was only trained for 1000 epochs. At this point, there is no clear advantage for QuinNet+FreeCG, and readers cannot ascertain whether extended training would improve QuinNet's performance.
>
> We appreciate your thoughtful comments. In fact, the results show that QuinNet+FreeCG trained for 1500 epochs already surpasses vanilla QuinNet. We provide the detailed numbers on the benchmarks as follows:
>
> | Model | Metric | Aspirin | Ethanol | Malonaldehyde | Naphthalene | Salicylic acid | Toluene | Uracil |
> |-------|---------|----------|----------|---------------|-------------|----------------|----------|---------|
> | QuinNet | Energy | 0.119 | 0.050 | 0.078 | 0.101 | **0.101** | 0.080 | 0.096 |
> | (from paper) | Force | 0.145 | 0.060 | 0.097 | 0.039 | 0.080 | 0.039 | 0.062 |
> | QuinNet | Energy | 0.132 | 0.052 | 0.076 | 0.109 | 0.106 | 0.081 | 0.099 |
> | (1500 epochs)| Force | 0.152 | 0.065 | 0.113 | 0.043 | 0.080 | 0.039 | 0.061 |
> | QuinNet + FreeCG | Energy | **0.113** | **0.048** | **0.073** | **0.094** | 0.103 | **0.078** | **0.094** |
> | (1500 epochs)| Force | **0.127** | **0.056** | **0.095** | **0.034** | **0.073** | **0.037** | **0.057** |
>
>
> It is noteworthy that FreeCG demonstrates its benefits to QuinNet by achieving superior performance with only 1/2 of the training time. We have added this comparative table to the revised paper. Thank you again for bringing this point to our attention.
>
> >For MILP, both model execution speed and memory usage are critically important. The proposed method seems to significantly increase the computational complexity of the model. It is crucial to include a comparison of the number of parameters and training speed.
>
> Thank you for this valuable suggestion! Here are the results:
>
> | | QuinNet+FreeCG | QuinNet | FreeCG | VisNet |
> |------------------|----------------|---------|---------|---------|
> | Number of parameters (M) | 9.4 | 9.1 | 10.4 | 9.8 |
> | Iterations per second (bs=32) | 4.19 | 4.45 | 3.62 | 3.95 |
>
>
> As shown above, FreeCG adds only a minor number of parameters compared to its baseline in both cases. Additionally, the training speed is comparable to the baselines. We would also like to point out that the other computational overhead is sufficiently low, as demonstrated in Figure 3, which displays the memory occupation and inference speed. Proposed Sparse Path and Group CG Transform reduce the computational overhead to minimum.
>
> We have incorporated all these results into our revised paper, which can be found in Tables 11 and 12. We sincerely hope that we have successfully addressed all your concerns. Should you have any further questions, we would be happy to discuss them. Again, thank you for your valuable feedback!

---

> > ### Comment · Reviewer_h6Dk · 2024-11-20
> >
> > Thank you for your response; it addressed my concerns. I have updated the score to 'accept.

---

> ### Author Response · Authors · 2024-11-20
> **Thank you for your positive response**
>
> Dear Reviewer,
>
> We are happy that we have resolved your concerns and we appreciate your responsible action in adjusting your score to reflect this. Your advice has been very helpful in improving our paper. We are glad that you acknowledge the value of FreeCG. We sincerely and humbly hope that FreeCG will be seen by the ICLR community and benefit further MLFFs research.
>
> Kind Regards,
>
> Anonymous Authors

---

### Official Review · Reviewer_o4xZ · 2024-10-30

**Soundness:** 2
**Presentation:** 2
**Contribution:** 2
**Rating:** 6
**Confidence:** 4

**Summary:**

The paper introduces a novel approach to Machine Learning Force Fields (MLFFs) that leverages permutation-invariant hidden features for efficient computation. These permutation-invariant features are aggregated over high-irreducible (high-irrep) edge features with attention weights, forming what is referred to as the “abstract edge” in the paper. The Clebsch-Gordan (CG) transformation is applied within each layer, and these permutation-invariant hidden features serve as the input of the kernel function for the CG transformation layer.

To improve the efficiency of the CG product, the authors propose a grouped CG transformation, which operates similarly to a group convolution. Additionally, to enhance the utility of the abstract edge, they introduce abstract edge shuffling and an Attention Enhancer mechanism.

**Strengths:**

- The paper includes extensive experiments in the MLFF domain, providing empirical solid support for the model’s performance.

- By aggregating high-irreducible (high-irrep) edge features with attention-weighted, permutation-invariant hidden features (abstract edges), the model achieves robust and flexible representations.

**Weaknesses:**

- The paper’s organization and readability could be improved. In particular, modified margins between paragraphs, figures, titles, and subtitles reduce overall readability.

- The paper does not adequately address  $\text{SO}(3)$ -equivariance, a critical concept in equivariant GNNs. Typically, in Clebsch-Gordan (CG) transformations, a radial distance-based kernel function is used to ensure  $\text{SO}(3)$ -equivariance. However, FreeCG may lack  $\text{SO}(3)$ -equivariance, as the aggregation of abstract edges is based on a weighted summation over components of each irreps. While this aspect is unclear, explicitly showing how the model achieves  $\text{SO}(3)$ -equivariance would enhance understanding. If the model does not fully satisfy  $\text{SO}(3)$ -equivariance, it would be helpful to justify using abstract edges within the CG transformation layer.

- The paper should include recent results in Table 2 to demonstrate the model’s state-of-the-art (SOTA) status. Specifically, adding comparisons with Graph ACE [1] and PONITA would strengthen the claim.

[1] Bochkarev, Anton, Yury Lysogorskiy, and Ralf Drautz. "Graph Atomic Cluster Expansion for Semilocal Interactions beyond Equivariant Message Passing." Physical Review X 14.2 (2024): 021036.

[2] Bekkers, Erik J., et al. "Fast, Expressive $\mathrm {SE}(n) $ Equivariant Networks through Weight-Sharing in Position-Orientation Space." The Twelfth International Conference on Learning Representations.

- In Equation (26), the model employs a summation over all permutations, which may constrain the model complexity to  O(N \times N!) . To fully discuss the model’s efficiency, scalability with respect to the number of atoms should also be considered.

### Minor Issues (Typos, Formatting, and Readability, etc..)
- Information about what kinds of loss are used is omitted.
- On page 5,  $N$  appears undefined and may be a typo; perhaps it should be  $Z$ .
- In Figure 3, the shaded regions impair readability; using a transparent background or adjusting the shading would improve clarity.
- Unnecessary line breaks are present in Appendix equations (7) and (15).
- Table 8 is unnecessarily expanded to fill the entire line width.

**Questions:**

- How does the model ensure the SO(3) equivariance?
- Why was the AIMD-Chig dataset chosen for memory usage and inference speed benchmarks?
- Why does abstract edge shuffling improve the model? A brief discussion on the role of shuffling is needed.

**Details Of Ethics Concerns:**

While this paper addresses machine learning force fields (MLFF), which are highly valuable in chemistry, physics, and materials science, the author does not mention any concerns about the potential misuse of this technology. For instance, there could be risks of malicious applications, such as creating chemical weapons. Although such scenarios are unlikely based on current understanding, I believe all researchers working in AI for scientific applications should remain vigilant about these possibilities, given that AI tools can be used without specialized knowledge.

---

> ### Author Response · Authors · 2024-11-14
> **Response to Reviewer o4xZ (1/2)**
>
> Dear Reviewer,
>
> We sincerely appreciate your comments on our paper.
>
> Before the full response to your review, we would like to first address one of your concern, which we believe is very important to be clarified.
>
> >The paper does not adequately address SO(3)-equivariance, a critical concept in equivariant GNNs. Typically, in Clebsch-Gordan (CG) transformations, a radial distance-based kernel function is used to ensure SO(3)-equivariance. However, FreeCG may lack SO(3)-equivariance, as the aggregation of abstract edges is based on a weighted summation over components of each irreps. While this aspect is unclear, explicitly showing how the model achieves SO(3)-equivariance would enhance understanding. If the model does not fully satisfy SO(3)-equivariance, it would be helpful to justify using abstract edges within the CG transformation layer.
>
> We respectfully suggest that this is a misunderstanding. The weighted sum of spherical irreps under the same $l$ is clearly SO(3)-equivariant (and if you add parity it is O(3)-equivariant, as we did in the paper). Here is the proof.
>
> *Proof. Let $v_i$ be the spherical irreps with order $l$, $g\in SO(3)$, $w_i$ the weight for $i$-th vector in the weighted sum, and $\rho_V$ the group homomorphism. We need to prove $\rho_V(g)\sum_i w_i v_i = \sum_i w_i \rho_{V_i}(g) v_i$. It holds if we can prove the following conditions 1) $\rho_{V_i}(g) w_i v_i = w_i \rho_{V_i}(g) v_i$ which is trivially true as $w_i$ is a scalar; 2) $\rho_{V_i} = \rho_{V_j}$ for arbitrary $i$ and $j$.*
>
> *To show 2) is golden, we need to show a) $V_i$ is an invariant subspace of SO(3), since the $\rho_{V_i}$ would be ill-defined if it is not the case, which is also trivially true because that is how spherical spaces are generated; b) $V_i = V_j$ for arbitrary $i$ and $j$. Here, $v_i$ are spherical irreps with same $l$. Spherical space of order $l$ is the same vector space spanned by spherical harmonics $Y^l_m$s. Spherical irreps with order $l$ are the vectors in the same order $l$ spherical space, so $V_i = V_j$ for arbitrary $i$ and $j$. Since all $V_i$s are the same we can simply denote each of them as $V$. Summarize over $i$ for both sides of $\rho_{V_i}(g) w_i v_i = w_i \rho_{V_i}(g) v_i$, we get $\rho_V(g)\sum_i w_i v_i = \sum_i w_i \rho_V(g) v_i$. This concludes our proof.  QED.*
>
> This proof can be well extended to other spaces if they satisfy the above arguments (eg. Cartesian space).
>
> This conclusion is taken for granted in most papers [1,2,3] in MLFFs and related areas without the explicit proof we have shown above. However, we are happy to add the proof to the revised paper per your suggestion to further enhance clarity (see Sec. A.5 in the current version). The only possible concern about equivariance/invariance is the permutation invariance as we generate abstract edges, which is introduced in Section A.4 in the appendix.
>
>
> To empirically verify this property, we provide the following code snippet:
>
> ```
> import e3nn.o3
> from e3nn.util.test import equivariance_error
>
> tp = e3nn.o3.FullyConnectedTensorProduct("2x0e + 3x1o", "2x0e + 3x1o", "2x1o")
>
>
> equivariance_error(
>     tp,
>     args_in=[tp.irreps_in1.randn(1, -1), tp.irreps_in2.randn(1, -1)],
>     irreps_in=[tp.irreps_in1, tp.irreps_in2],
>     irreps_out=[tp.irreps_out]
> )
> ```
> This snippet of code weighted sums the spherical irreps under the same $l$, and obtains the error of equivariance test. You could run this code, and you would see that the error is around $10^{-8}$, where the only source of error is from computational stability (floating-point precision). This level of error is common and acceptable for equivariant neural networks. We have also empirically tested the equivariance error of our model. It is around $1.41\times 10^{-6}$ for energy prediction, and around $4.92\times 10^{-6}$ for force prediction.
>
> We hope this can address your concern about the SO(3)-equivariance. If you have any further questions on this point, please feel free to ask. We are happy to provide clarification.
>
> [1] Batzner, S., Musaelian, A., Sun, L., Geiger, M., Mailoa, J. P., Kornbluth, M., ... & Kozinsky, B. (2022). E (3)-equivariant graph neural networks for data-efficient and accurate interatomic potentials. Nature communications, 13(1), 2453.
>
> [2] Simeon, G., & De Fabritiis, G. (2024). Tensornet: Cartesian tensor representations for efficient learning of molecular potentials. Advances in Neural Information Processing Systems, 36.
>
> [3] Musaelian, A., Batzner, S., Johansson, A., Sun, L., Owen, C. J., Kornbluth, M., & Kozinsky, B. (2023). Learning local equivariant representations for large-scale atomistic dynamics. Nature Communications, 14(1), 579.

---

> ### Author Response · Authors · 2024-11-19
> **Response to Reviewer o4xZ (2/2)**
>
> Dear reviewer,
>
> Thanks for your patience! We have spent time carefully addressing each of your concerns. This response is to the other concerns you raised.
>
>  > The paper’s organization and readability could be improved. In particular, modified margins between paragraphs, figures, titles, and subtitles reduce overall readability.
>
> Thanks for your advice! We have adjusted all the margins to the original form, making it neat and elegant, and rearranged the context to fit the length. Please refer to the current version of our paper.
>
> > The paper should include recent results in Table 2 to demonstrate the model’s state-of-the-art (SOTA) status. Specifically, adding comparisons with Graph ACE [1] and PONITA would strengthen the claim.
>
> Thank you for your advice to further enhance our paper. We have added the results of the two papers you mentioned, and also added the citations to give credit to their works.
>
> > In Equation (26), the model employs a summation over all permutations, which may constrain the model complexity to O(N \times N!) . To fully discuss the model’s efficiency, scalability with respect to the number of atoms should also be considered.
>
> Unfortunately, this could be another misunderstanding. The permutation is implicitly included in the aggregation operation. Recall the equation we wrote:
>
> $\hat{E}^L_{i}=\sum_{j\in\mathcal{N}(i)}\hat{E}^L_{j\mapsto i} = \sum_{p\in P} \sum_{j\in\mathcal{N}(i)}\frac{\mathcal{P}(p)\hat{E}^L_{j\mapsto i}}{{\rm Card}(P)}$
>
> This equation means that we actually **conduct the middle part of this equation**, ie. $\sum_{j\in\mathcal{N}(i)}\hat{E}^L_{j\mapsto i}$, and **it is equivalent to we conduct the last part of the equation**, ie. $\sum_{p\in P} \sum_{j\in\mathcal{N}(i)}\frac{\mathcal{P}(p)\hat{E}^L_{j\mapsto i}}{{\rm Card}(P)}$. The last part demonstrates that our operation is permutation equivariant. We further elaborate this point in L970-971 to further enhance clarity.
>
> > Information about what kinds of loss are used is omitted.
>
> Thanks for your catch! Please refer to L430 in our revised paper.
>
> > On page 5, N appears undefined and may be a typo; perhaps it should be Z.
>
> You are definitely right. We have corrected all the typo for $Z$. Thanks for your catch!
>
> > In Figure 3, the shaded regions impair readability; using a transparent background or adjusting the shading would improve clarity.
>
> Honestly, we did not see any shaded region in Figure 3 (probably due to the PDF reader or stuff). We have set the background of this figure to transparent as you suggest (now Figure 5). Please see if the current figure looks right to you?
>
> > Unnecessary line breaks are present in Appendix equations (7) and (15).
>
> We have deleted the line breaks.
>
> > Table 8 is unnecessarily expanded to fill the entire line width.
>
> We have adjusted the size of Table 8.
>
> > How does the model ensure the SO(3) equivariance?
>
> Please refer to our previous response (1/2).
>
> > Why was the AIMD-Chig dataset chosen for memory usage and inference speed benchmarks?
>
> It contains mini-protein which is more challenging for inference speed and memory usage. Previous works (eg. VisNet) have benchmarked the efficiency on it. We have added this in L496-497.
>
> > Why does abstract edge shuffling improve the model? A brief discussion on the role of shuffling is needed.
>
> If we omit the shuffling operation, the information in the abstract edges would only be exchanged within the group, which decreases the capacity of the model. We have added the discussion in L356-358.
>
> > While this paper addresses machine learning force fields (MLFF), which are highly valuable in chemistry, physics, and materials science, the author does not mention any concerns about the potential misuse of this technology. For instance, there could be risks of malicious applications, such as creating chemical weapons. Although such scenarios are unlikely based on current understanding, I believe all researchers working in AI for scientific applications should remain vigilant about these possibilities, given that AI tools can be used without specialized knowledge.
>
> We completely agree that the limitation discussion is needed in this case. We have added the content in the Conclusion section, where it reads:
>
> *Meanwhile, it is crucial to ensure that the use of MLFFs is strictly regulated and controlled to prevent their misuse for illegal purposes, such as the development and deployment of chemical weapons. Furthermore, global cooperation and information sharing between academic community and industry are key to identifying and mitigating potential threats in a timely manner.*
>
> We thanks the effort you put to review our paper. We have also put tremendous time to address each of your questions raised. Hope we have clarified all of your concerns! We really appreciate it if we could hear from you whether you are satisfied with our response.

---

> ### Author Response · Authors · 2024-11-21
> **Additional comment**
>
> Dear Reviewer,
>
> We are sorry for the disturbance and we know you must be quite busy this time. Could you please let us know if you are satisfied with our response and the revisions to the paper? Does the previously unclear point now make sense to you? Since there is some potential misunderstanding so we are eager to know whether they have been properly clarified. Thanks in advance for your time and efforts!
>
> Sincerely,
>
> Anonymous authors

---

> > ### Comment · Reviewer_o4xZ · 2024-11-22
> >
> > Thank you for your thoughtful response and for addressing my concerns. I appreciate the efforts you have made to clarify the previously unclear points and enhance the paper's readability.
> >
> > I will need some additional time to carefully verify your answers and review the revised version in detail. That said, the improvements in readability are indeed valuable, and I am inclined to consider raising my score based on these enhancements after further evaluation.
> >
> > Thank you again for your efforts and your patience.

---

> > > ### Author Response · Authors · 2024-11-22
> > > **Thank you for your responsible reply**
> > >
> > > Dear Reviewer,
> > >
> > > We sincerely appreciate your prompt response and fully respect your need for additional time to carefully review our answers and revisions. Should any questions arise during your consideration, please feel free to reach out. We would be delighted to provide further clarification! Once again, we thank you for your valuable feedback to enhance our paper's presentation.
> > >
> > > Kind Regards,
> > >
> > > Authors

---

> > > ### Author Response · Authors · 2024-11-22
> > > **A potential misunderstanding from our side**
> > >
> > > We apologize for any inconvenience. We've been busy with experiments lately, but your comment prompted us to further reassess our response. Upon review, we realized there might have been a misunderstanding regarding one of your questions:
> > >
> > > > In Figure 3, the shaded regions impair readability; using a transparent background or adjusting the shading would improve clarity.
> > >
> > > We have thought that you mean the background of the figure is shaded as it sometimes occurs for non-transparent background. We just realize that most likely what you meant was the orange region instead. We sincerely apologize if this was the case. We now assume that it was our misunderstanding (we believe, most likely), and change the figure again accordingly.
> > >
> > > Thank you for your understanding!

---

> ### Comment · Reviewer_o4xZ · 2024-11-24
>
> Thank you for your clarification.
>
> First of all, I acknowledge that I initially overlooked the simple way of proof of equivariance, which was clarified in your first comment. My overall impression of the paper has improved, particularly with the corrections to the marginal issues, typos, and the updated figure. I believe this simple yet effective approach to handling equivariance via attention mechanism is a meaningful contribution to the community, even though it does not achieve state-of-the-art results. Based on this, I have decided to raise my overall score from 3 to 6 and the presentation score to 2.
>
> Furthermore, I would like to suggest further clarification, regarding how the model and concept ensure equivariance. The focus should be on the overall idea architecture and flows, rather than the implementation details, such as the linear combination of the irreducible representations. Additionally, while I recommended including an ethics statement following [ICLR guidelines](https://iclr.cc/Conferences/2024/AuthorGuid), it is acceptable to move this section to the appendix or place it after the conclusion if you require more space in the main text.
>
> Thank you again for your thoughtful response and revisions.

---

> > ### Author Response · Authors · 2024-11-25
> >
> > Dear Reviewer,
> >
> > Thank you for your positive feedback! We have carefully revised the paper according to your latest suggestions and reminders. Please find the updated version attached. Specifically, we have outlined the core concept of preserving both O(3) and permutation equivariance in L406-412 and included an ethics statement following the conclusion.
> >
> > We sincerely appreciate your valuable efforts throughout the review process.
> >
> > Kind Regards,
> >
> > Authors

---

### Official Review · Reviewer_dHAu · 2024-11-01

**Soundness:** 2
**Presentation:** 2
**Contribution:** 3
**Rating:** 6
**Confidence:** 3

**Summary:**

This paper proposes an approach to address the computational inefficiency of Clebsch-Gordan (CG) transforms in rotation-translation equivariant graph neural networks (EGNNs). Leveraging the invariance transitivity property, the proposed method, FreeCG, implements the CG transform layer on permutation-invariant abstract edges, enabling a more flexible layer design without compromising permutation equivariance. Additional architectural modifications are introduced to enhance model efficiency, with extensive empirical results demonstrating FreeCG’s performance.

**Strengths:**

1. FreeCG could reduce the computational overhead of CG transforms and enhance expressivity by incorporating several architectural improvements.
2. FreeCG achieves competitive or superior performance across multiple benchmarks and demonstrates compatibility with various EGNN architectures.

**Weaknesses:**

1. FreeCG’s approach to freeing CG transform space with abstract edges resembles the customized tensor product mechanism used in Allegro (see Eq. 13 in [1]). Allegro applies CG transforms to geometric features of a pair (similar to FreeCG’s node features) and environment embeddings of the pair (similar to FreeCG’s abstract edges from neighbors). While FreeCG focuses on atom-wise message passing with greater interaction between atom features and abstract edges, further clarification on the advantages of FreeCG over Allegro would strengthen the technical contribution.
2. Introducing sparse paths may come with a trade-off in expressivity. The authors should provide an ablation study to analyze the impact of sparse paths on computational efficiency and performance.
3. Results in Table 5 are somewhat unclear. If the final FreeCG model uses the best-performing modules, it should have a group number of 32, which conflicts with the number 8 reported in Table 7. Additionally, the ablations indicate that the primary performance improvement is due to increased group numbers, suggesting that the other modules’ contributions may be limited.
4. Certain aspects of the methodology require additional clarity: a) Some notations are under-defined, e.g., $N$ in line 227, $\bar{E_i^L}$, and $d\bar{E_i^{L+1}}$ in line 277. b) The relationship between $d\bar{E_i^{L+1}}$ and $\bar{E_i^{L+1}}$ in line 277 is unclear, and more details are needed on how $\bar{E_i^L}$ is constructed in the first layer.

[1] Musaelian, A., Batzner, S., Johansson, A. et al. Learning local equivariant representations for large-scale atomistic dynamics. Nat Commun 14, 579 (2023).

**Questions:**

1. For abstract edge shuffling, did the authors try using a linear layer to mix different channels of $\hat{E}_i^L$? Linear mixing could enhance information exchange effectively in this case.
2. Did the authors consider applying FreeCG to more representative tensor product-based EGNNs? This could further validate the method’s efficacy.

---

> ### Author Response · Authors · 2024-11-19
> **Response to Reviewer dHAu**
>
> Dear Reviewer,
>
> We truly appreciate your professional suggestions to help our work better pronounced.
>
> > FreeCG’s approach to freeing CG transform space with abstract edges resembles the customized tensor product mechanism used in Allegro (see Eq. 13 in [1]). Allegro applies CG transforms to geometric features of a pair (similar to FreeCG’s node features) and environment embeddings of the pair (similar to FreeCG’s abstract edges from neighbors). While FreeCG focuses on atom-wise message passing with greater interaction between atom features and abstract edges, further clarification on the advantages of FreeCG over Allegro would strengthen the technical contribution.
>
> We must say the comparison you suggest is a very good point. **Allegro constructs high order geometric features with pair-wise edge CG transform, which is similar to FreeCG at this point. However, Allegro requires to compute the same CG transform for each pair of edges in the same manner to maintain permutation equivariance, while FreeCG is capable of designing different CG transform for each pair of abstract edges because of their permutation invariance. The atom-centric style and the rich representation of abstract edges together makes stronger expressivity of FreeCG.** We have added the above discussion to L126-131.
>
> > Introducing sparse paths may come with a trade-off in expressivity. The authors should provide an ablation study to analyze the impact of sparse paths on computational efficiency and performance.
>
> We find this advice very valuable. We have added additional ablation study on sparse paths, as shown below:
>
> |                   | Force MAE | Energy MAE | Train. speed | Num. of param. |
> |-------------------|-----------|------------|--------------|----------------|
> | **Sparse path**   | 0.122     | 0.110      | 3.62         | 10.4M          |
> | **Full path**     | 0.118     | 0.106      | 0.79         | 43.8M          |
>
> The results show that sparse path is indeed a better trade-off than full path. We have added this table to our revised paper as Table 8. The inference speed is already shown in Fig. 3.
>
>
> > Results in Table 5 are somewhat unclear. If the final FreeCG model uses the best-performing modules, it should have a group number of 32, which conflicts with the number 8 reported in Table 7. Additionally, the ablations indicate that the primary performance improvement is due to increased group numbers, suggesting that the other modules’ contributions may be limited.
>
> We apologize for this typo. The number in Table 7 should be 32. Thanks for the good catch! For your second concern, please note that group shuffling and attention enhancer perform sufficiently good given that group CG transform have already decrease the MAE to a very low level. The experiment on sparse paths as you suggest (thanks again) has also proved its efficacy.
>
> >Certain aspects of the methodology require additional clarity: a) Some notations are under-defined, e.g., $N$ in line 227, $\overline{E}$, and $d\overline{E}$ in line 277. b) The relationship between $\overline{E}$ and $d\overline{E}$ in line 277 is unclear, and more details are needed on how $\overline{E}$ is constructed in the first layer.
>
>
> $N$ is actually $Z$ which are atom types. Thanks for your catch! We have added description of $\overline{E}$ and $d\overline{E}$ in L295-296, which reads **"and $\overline{E}$ represents propagated abstract edges in contrast to the temporary one $\hat{E}$, where the former is updated by $d\overline{E}$ in each layer, derived from the temporary abstract edge $\hat{E}$."**. The way we initialize $\overline{E}$ in the first layer is mentioned in L755, **"We also maintain zero-initialized abstract edges $\overline{E}^{L=0}_i={0}$ for each node to be updated in the following layers."**
>
>
>
> > For abstract edge shuffling, did the authors try using a linear layer to mix different channels of $\hat{E}$? Linear mixing could enhance information exchange effectively in this case.
>
> Thank you for your suggestion. We have run a new model based on your alternative design. It shows similar results on Aspirin in MD17 where the energy MAE is 0.113 and force MAE 0.119. We would say that the force prediction is indeed improved marginally, but the number of parameters are raised from 10.4M to 11.0M for the multiple 256x256 weights added (to reduce the cost, a bottleneck structure would be a potential way out here, but we are not so confident about the performance). Overall, it remains somehow not worthy regarding the trade-off. But we do appreciate your suggestion and this architecture change is intuitive, but unfortunately, it does not appear the way we expect.
>
>
> > Did the authors consider applying FreeCG to more representative tensor product-based EGNNs? This could further validate the method’s efficacy.
>
> Yes. We plan to add FreeCG to TensorNet to see the outcome. The rebuttal period is too tight to perform this experiment, but we will leave this after the rebuttal.

---

> ### Author Response · Authors · 2024-11-20
> **Response to Reviewer dHAu (cont.)**
>
> > Did the authors consider applying FreeCG to more representative tensor product-based EGNNs? This could further validate the method’s efficacy.
>
> We apologize that we have overlooked the "tensor product-based" part in the previous response. In this case, we plan to add FreeCG to TensorNet [1] to see the performance.
>
> Also, could you let us know whether you are satisfied with our response when you have the time? That would mean a lot to us. Thank you!
>
> [1] Simeon, G., & De Fabritiis, G. (2024). Tensornet: Cartesian tensor representations for efficient learning of molecular potentials. Advances in Neural Information Processing Systems, 36.

---

> > ### Comment · Reviewer_dHAu · 2024-11-25
> >
> > Hi Authors,
> >
> > Thank you for the clarification. The response has addressed my concerns well, and I will raise my score.

---

> > > ### Author Response · Authors · 2024-11-25
> > >
> > > Dear Reviewer,
> > >
> > > Thank you for your kind feedback! We're thrilled to know we successfully addressed your concerns. Your comments and suggestions are invaluable to us—thank you for helping us improve!
> > >
> > > Kind Regards,
> > >
> > > Authors

---

### Official Review · Reviewer_mCwJ · 2024-11-04

**Soundness:** 2
**Presentation:** 2
**Contribution:** 2
**Rating:** 6
**Confidence:** 4

**Summary:**

This paper proposes FreeCG, a method that implements the CG transform layer on the permutation-invariant abstract edges, which allows complete freedom in the design of the layer without compromising the overall permutation equivariance. This can greatly improve the model’s expressiveness and decrease the computational demands.

**Strengths:**

- This paper uses invariance transitivity with permutation-invariant abstract edges to solve the narrowness design space of CG transform.
- This work further proposes a FreeCG model that contains Group CG transform with sparse path, abstract edges shuffling, and Attention enhancer to improve the representation power and efficiency.
- The model shows good performance on several small molecule datasets.

**Weaknesses:**

- In Table 2, FreeCG is not effective in energy prediction on the rMD17 dataset. Could authors elaborate more on this?
- In Table 4, other competitive baselines such as equiformerV2 should be included for a comprehensive comparison.
- The datasets used in this paper are quite small and the results are sometimes not robust to evaluate the model performance, such as QM9. I would like to see the force and energy prediction performance on a large dataset such as OC20, to better understand the effective and efficiency of the FreeCG.

**Questions:**

- Please refer to the weakness.

---

> ### Author Response · Authors · 2024-11-19
> **Response to Reviewer mCwJ**
>
> Dear Reviewer,
>
> We sincerely appreciate your devotion to the valuable comments, and we are happy to address them. Meanwhile, we apologize in advance that we compare with some other works for MLFFs, but we truly respect their works and believe they are very robust and fundamental contributions. We merely would like to show that our work meets the high standard.
>
> > In Table 2, FreeCG is not effective in energy prediction on the rMD17 dataset. Could authors elaborate more on this?
>
> We have tuned the loss weight more on the energy side (0.9*force + 0.1*energy now), and it shows that it can perform SoTA for energy under this setting without affecting the force prediction.
>
> |         |        | UNiTE | GemNet | NequIP |  MACE  | Allegro | BOTNet | VisNet | QuinNet | ACE$^{g}$  | P$\Theta$NITA |   FreeCG   |
> |---------|--------|:-----:|:------:|:------:|:------:|:-------:|:------:|:------:|:-------:|:----------:|:-------:|:----------:|
> | Aspirin | Energy | 0.055 |    -   | 0.0530 | 0.0507 |  0.0530 | 0.0530 | 0.0445 |  0.0486 | 0.0392 | 0.0392 | **0.0374** |
> |         | Force  | 0.175 | 0.2191 | 0.1891 | 0.1522 |  0.1684 | 0.1900 | 0.1520 |  0.1429 | 0.1407  | 0.1338 | **0.1225** |
>
> The rest of the experiments are still running. From what we see from Aspirin we believe that this strategy has very strong potentiality towards a SOTA energy prediction. We have also added this table to our revised paper as Table 10. Meanwhile, we believe that in MLFFs, it is more reasonable to focus on the force prediction results, as the downstream task is usually for molecular dynamics. Some works did not even report the energy prediction results [1] or underperforms others on energy prediction while being SOTA on force prediction (eg. QuinNet on rMD17 [2] and Allergo on 3BPA [3]). But they are all robust and valuable methods while they focus mainly on the force prediction side. However, we agree that a good performance of energy prediction can further prove the capacity of the model. So we will rerun the rest of the molecules under new loss weights and early stopping strategies.
>
> > In Table 4, other competitive baselines such as equiformerV2 should be included for a comprehensive comparison.
>
> We have added the results per your suggestion and added the citation to acknowledge the credit. Please see the revised Table 6.
>
> > The datasets used in this paper are quite small and the results are sometimes not robust to evaluate the model performance, such as QM9. I would like to see the force and energy prediction performance on a large dataset such as OC20, to better understand the effective and efficiency of the FreeCG.
>
> Thanks for your advice. But in the current state, it is unrealistic to finish the OC20 training in the discussion period. We hope that you could understand that. We are very willing to run the experiments but it might be later than the deadline of the discussion period. In fact, we have already run extensive experiments from diverse molecules and trajectories: MD17, rMD17, MD22, QM9, Chignolin, and perodic systems, including water and LiPS. We devoted very long time and efforts to the whole set of experiments and surpasses the most published, outstanding works in this line. It is also noteworthy that, the mean system size in OC20 is 77.75, while the atom numbers of supermolecules in MD22 ranges from 42 to 370, and the size of Chignolin is 166. We thus believe they are enough to demonstrate the capacity of FreeCG to predict diverse molecules. There are many papers reporting the results on the benchmarks we chose but without OC20 results [2,3,4]. There are also works reporting OC20 but not giving as much results on our experiments conducted. Again, we are very willing to add OC20 experiments after the discussion period considering the tight discussion time period, but we also believe that the current experimental load is adequate to demonstrate the efficacy of FreeCG.
>
> We know that you intend to improve our works through some advices which we sincerely appreciate, and hope our response addresses your concerns!
>
> [1] Gasteiger, J., Becker, F., & Günnemann, S. (2021). Gemnet: Universal directional graph neural networks for molecules. Advances in Neural Information Processing Systems, 34, 6790-6802.
>
> [2] Wang, Z., Liu, G., Zhou, Y., Wang, T., & Shao, B. (2023). QuinNet: efficiently incorporating quintuple interactions into geometric deep learning force fields. In Proceedings of the 37th International Conference on Neural Information Processing Systems. 77043-77055.
>
> [3] Musaelian, A., Batzner, S., Johansson, A., Sun, L., Owen, C. J., Kornbluth, M., & Kozinsky, B. (2023). Learning local equivariant representations for large-scale atomistic dynamics. Nature Communications, 14(1), 579.
>
> [4] Batzner, S., Musaelian, A., Sun, L., Geiger, M., Mailoa, J. P., Kornbluth, M., ... & Kozinsky, B. (2022). E (3)-equivariant graph neural networks for data-efficient and accurate interatomic potentials. Nature communications, 13(1), 2453.

---

> ### Comment · Reviewer_mCwJ · 2024-11-25
>
> Thanks for the effort. I will raise my score, and I am looking forward to the OC20 results!

---

> > ### Author Response · Authors · 2024-11-25
> > **Thank you for your positive feedback**
> >
> > Dear Reviewer,
> >
> > We appreciate your positive feedback. We will complete the experiments on OC20 s2ef after the rebuttal. Thank you once again for your valuable suggestions to further enhance our work!
> >
> > Kind Regards,
> >
> > Authors

---

### Author Response · Authors · 2024-11-19
**General response**

Dear Reviewers and ACs,

We would like to sincerely appreciate the time and effort you devoted to our paper. We are glad that all reviewers acknowledge the novelty and efficacy of our FreeCG for machine learning force fields. The extensiveness of the experiments that took months for us to complete are also widely acknowledged (Reviewer dHAu, o4xZ, and h6Dk).

We have carefully addressed each of reviewer's concerns, and added several experiments and results suggested by reviewers at this stage to our revised paper. Here are the following details:

 loss weight of 0.9\*force MAE+0.1\*energy MAE
|         |        | UNiTE | GemNet | NequIP |  MACE  | Allegro | BOTNet | VisNet | QuinNet | ACE$^{g}$  | P$\Theta$NITA |   FreeCG   |
|---------|--------|:-----:|:------:|:------:|:------:|:-------:|:------:|:------:|:-------:|:----------:|:-------:|:----------:|
| Aspirin | Energy | 0.055 |    -   | 0.0530 | 0.0507 |  0.0530 | 0.0530 | 0.0445 |  0.0486 | 0.0392 | 0.0392 | **0.0374** |
|         | Force  | 0.175 | 0.2191 | 0.1891 | 0.1522 |  0.1684 | 0.1900 | 0.1520 |  0.1429 | 0.1407  | 0.1338 | **0.1225** |

Ablation on sparse path
|                   | Force MAE | Energy MAE | Train. speed | Num. of param. |
|-------------------|-----------|------------|--------------|----------------|
| **Sparse path**   | 0.122     | 0.110      | 3.62         | 10.4M          |
| **Full path**     | 0.118     | 0.106      | 0.79         | 43.8M          |

QuinNet equipped with FreeCG trained for 1500 epochs
| Model | Metric | Aspirin | Ethanol | Malonaldehyde | Naphthalene | Salicylic acid | Toluene | Uracil |
|-------|---------|----------|----------|---------------|-------------|----------------|----------|---------|
| QuinNet | Energy | 0.119 | 0.050 | 0.078 | 0.101 | **0.101** | 0.080 | 0.096 |
| (from paper) | Force | 0.145 | 0.060 | 0.097 | 0.039 | 0.080 | 0.039 | 0.062 |
| QuinNet | Energy | 0.132 | 0.052 | 0.076 | 0.109 | 0.106 | 0.081 | 0.099 |
| (1500 epochs)| Force | 0.152 | 0.065 | 0.113 | 0.043 | 0.080 | 0.039 | 0.061 |
| QuinNet + FreeCG | Energy | **0.113** | **0.048** | **0.073** | **0.094** | 0.103 | **0.078** | **0.094** |
| (1500 epochs)| Force | **0.127** | **0.056** | **0.095** | **0.034** | **0.073** | **0.037** | **0.057** |



A major issue about the equivariance of FreeCG has been addressed in the response to Reviewer o4xZ who suspected that irreps under the same order $l$ cannot be weighted sum. We have clarified that it was a misunderstanding and presented rigorous proof and experimental results to show the point. That being said, it is common to suspect something or make mistake (we deeply apologize if irreps under same $l$ cannot be weighted summed up, but each of the authors believe it can be and there are so many papers could be taken as evidences) as we did sometimes, and we still greatly appreciate the other comments on the writing details from Reviewer o4xZ and carefully addressed them in one-to-one response. We also agree that potentiality for chemical weapon is worth mentioning, as we added the content in the revised paper. Here we provide another snippet to show the equivariance:

```
from e3nn import o3
import torch

rot = o3.wigner_D(torch.tensor(2),torch.tensor(0.1),torch.tensor(0.2),torch.tensor(0.3))

a = torch.rand(5)
b = torch.rand(5)

2.5*(rot@a) + 1.4*(rot@b) - rot@(2.5*(a) + 1.4*(b))
```
The results would be something like:
```
tensor([-2.3842e-08,  0.0000e+00,  0.0000e+00,  1.1921e-08,  1.7881e-08])
```
One could arbitrarily change 2.5 and 1.4 as the weights. We pick a non-trivial order $l=2$. The error in this case is in 10$^{-8}$ level, which is enough for claiming the equivariance.


Overall, we humbly suggest FreeCG is a substantial contribution to the MLFFs community. It is with reasonable motivation, novel technical contribution, and significant experimental SOTA on extensive test benchmarks. We respectfully think we have solved major concerns raised by reviewers, and hope that one could see the merits it has.

---

### Meta-Review · Area_Chair_DJnX · 2024-12-20

**Metareview:**

This paper introduces a clever, highly expressive neural parameterization of force fields which obeys the requisite equivariances without increasing computational complexity.  The new parameterization is then applied to a suite of challenging molecular benchmarks where it achieves strong results across the board, often improving SOTA by quite a margin.

The reviewers initially criticised clarity and the scope of experiments and comparisons, but in the rebuttal phase the authors put in a massive effort to alleviate these criticisms, run many new experiments, and clarify things. There has been some concerns about SO(3) equivariance by o4xZ but it is clear that this is satisfied and the authors have argued with both derivations and numerics.

Overall the rebuttal phase made it clear that this is a solid, original contribution to the field of machine learned force fields and with the all the improvements that the authors made and the positive assessments from the reviewers, I can only recommend acceptance.

**Additional Comments On Reviewer Discussion:**

The discussion was respectful and productive. mCwJ suggested that the datasets are small and there are baselines missing but the authors argued convincingly that it's not really the case (and I agree). The authors nonetheless ran new experiments and tested new baselines. dHAu pointed out similarities with Allegro which and had concerns about expressivity with sparse paths. After the authors addressed these points. o4xZ worries about SO(3) / O(3) equivariance but that is clearly not a concern—authors responded with code and derivations. h6Dk commented that the QuinNet results are lackluster but the authors convincingly argued that the opposite is true, so h6Dk raised their score.

My decision is based on a careful analysis of the discussions, and in particular the fact the proposed parameterization is indeed original and useful and the problem is of high importance. I thought that already the initial experiments were impressive. All reviewers recommend acceptance.

---

### Decision · Program_Chairs · 2025-01-22

Accept (Poster)